# Optimal Sub-data Selection for Nonparametric Function Estimation in Kernel Learning with Large-scale Data

## Abstract

This paper considers estimating nonparametric functions in a reproducing kernel Hilbert space (RKHS) for kernel learning problems with large-scale data. Kernel learning with large-scale data is computationally intensive, particularly due to the high cost and complexity of tuning parameter selection. Existing sampling methods for scalable kernel learning, such as the leverage score-based sampling method and its variants, are designed to sketch the kernel matrix to minimize the expected global (in-sample or out-of-sample) prediction error. In complement to existing methods, this paper proposes an optimal informative sampling method to estimate nonparametric functions pointwisely when the subsample size is potentially small. Our method is tailored for scenarios where computational resources are limited, yet accurate pointwise prediction at each test location is desired. It aims to select an informative sub-data, which is different from sketch methods that aim to select the columns of a kernel matrix. Theoretical studies compare the efficiency of the proposed method to that based on the full data with optimally selected tuning parameters. Furthermore, integrating our sub-data selection with sketching methods can improve computation time of full-data based sketching methods while maintaining the same statistical efficiency. Numerical experiments demonstrate the statistical and computational efficiency of the proposed method.

## 1 Introduction

Large-scale data sets that have a large number of records with a great variety of resources are increasingly common in many applications such as genomics and genetics, neuro-imaging, and finance. While the increasing amount of data size brings tremendous potential for discoveries and makes it possible to fit complex models such as deep neural network models, it also brings tremendous challenges to many existing algorithms to process and analyze data quickly and efficiently. This paper is on large-scale data sets with large sample sizes. With increasing complexity and heterogeneity, nonparametric models are reliable and realistic because of their flexibility in the assumption and structure of the model. This paper focuses on nonparametric regularizations in an RKHS or the so-called kernel machine methods (Wahba, 1990; Wang, 2011; Gu, 2013; Liu et al., 2007).

The statistical properties of the kernel machine methods have been well-documented. However, the computation of the kernel machine method can be challenging for large-scale data sets. It is well known that the computational cost is at the order of $O(N^3)$ using a direct computation, where $N$ is the sample size. To address the computational challenge, Zhang et al. (2015) developed a divide-and-conquer approach. The divide-and-conquer approach (Chen et al., 2021; Li et al., 2013) has been one of the most frequently used strategies. It first breaks down large-scale data into independent processable subsets, sending them to distributed machines for processing to obtain intermediate results, and these intermediate results are merged into final results. Besides the divide-and-conquer approach, there have been various approximation methods developed in the literature, including random Fourier features (Yang et al., 2012; Rahimi & Recht, 2007), the Nyström method (Williams & Seeger, 2001), FALKON (Rudi et al., 2017; Meanti et al., 2020) and EigenPro method (MA & Belkin, 2017; Abedsoltan et al., 2023). While random Fourier features approximate the functions in an RKHS using a smaller number of randomly sampled basis functions, Nyström method applies the idea of sketch to replace the empirical kernel matrix by a much smaller matrix with subsampled

columns. The FALKON and EigenPro method combine the preconditioning idea and the random projection idea to speed up the computation.

The article aims to develop a sub-data selection-based method for nonparametric function estimation in an RKHS to achieve the best statistical efficiency when the computational resource is limited (Yao & Wang, 2021). The goal of our sub-data selection is to select the most informative subset of observations, which is different from the existing sketch methods based on subsampling methods, such as leverage score-based sampling(Alaoui & Mahoney, 2015; Rudi et al., 2015; 2018), which have been developed to subsample columns of a kernel matrix. There exist abundant sub-data selection approaches for data generated from parametric models. For example, Drineas et al. (2006); Mahoney (2011) considered leverage sampling and Wang et al. (2019a) proposed an information-based optimal subdata selection (IBOSS) to find subdata with the maximal information matrix under the D- optimality for linear regression models. Ma & Sun (2015) developed local case-control sampling and Cheng et al. (2020) generalized the idea of IBOSS for logistic regression models. In addition, subdata selection approaches have been proposed for generalized linear models (Ai et al., 2018) and quantile regression (Wang & Ma, 2020). A comprehensive review of these existing subdata selection methods for parametric models may be found in Yao & Wang (2021); Chang (2024). However, there is limited research on subdata selection methods targeting on nonparametric function estimation. Recently, Chang (2024) developed a stratified subsampling approach for a supervised learning in a nonparametric model setting. It is based on a partitioning estimate that is similar to the regression tree or Nadaraya-Watson kernel estimator, which is different from the kernel machine method discussed in this paper.

Inspired by the existing sub-data selection methods, we propose a new sub-data selection methodology to estimate nonparametric functions in an RKHS. We first apply a clustering method such as k-means or other clustering algorithms to select representative data points. The nonparametric function values in each cluster are roughly approximated by the functional values at representative data points. To decide the sampling weights for each cluster, we minimize the MSE of the nonparametric function estimator that is constructed based on the selected representative data points. However, the sampling weights depend on a tuning parameter which is unknown. To address this issue, we adopted a one-step iteration procedure to choose a tuning parameter using representative data points via the BIC criterion or cross-validation. The optimal weights depend on the cluster centers decided by the K-means algorithm, the chosen kernel function, and the selected tuning parameter. After selecting multiple sub-datasets using the optimal weights, we apply the kernel machine method to each selected sub-dataset and aggregate the estimators to obtain the final estimate. We further show that the sub-data selection method can be integrated with sketching methods such as FALKON to improve the computation time for pointwise estimation while maintaining the same statistical efficiency.

From a theoretical perspective, the proposed method is designed to select a sub-sample (with a fixed sample size) to minimize the expected prediction error at every test data point, hence it minimizes the expected prediction error for every test data set. It is different from existing research for sketch method in kernel learning, which has established its optimality to minimize the expected global prediction error. For example, Musco & Musco (2017) considers in-sample prediction error and Rudi et al. (2015; 2018); Alaoui & Mahoney (2015) consider the expected global generalization (out-of-sample) prediction error. While these results are interesting and important, they do not directly guarantee generalization performance for every test data point. Our contribution is to establish the rate of convergence of the proposed estimator, and demonstrate its advantage in improving the convergence rate of the RKHS estimator based on subsamples obtained from a Simple Random Sampling (SRS). The results are interesting since they confirm that the proposed sub-data selection is informative in making use of the full data to improve the prediction error.

From a numerical perspective, our numerical results show that our proposed method is computationally efficient when compared with a Simple Random Sample (SRS) approach and maintains estimation accuracy when compared with the full data approach. The integrated mean squared errors of our proposed method are comparable to that of using full data while computational time is as good as the SRS based approach. More importantly, the proposed approach achieves a good balance between statistical efficiency and computational efficiency when compared with the full data-based and the SRS-based approaches. In addition, we investigate a combination of proposed method with sketching methods (e.g., FALKON) to improve computation time of existing sketching methods. The proposed method has shown better MSE in out-of-sample prediction than some existing sketch algorithms, including Nyström (Williams & Seeger, 2001) and FALKON (Rudi et al., 2017; Meanti

et al., 2020). We also compare the performance of the proposed method with these sketch algorithms in the YearPredictionMSE data set.

This manuscript is organized in the following way. In Section 2, we provide the basic framework, an introduction of regularization in RKHS with full data sets, and the proposed method for regularization in RKHS with large-scale data, with the details of our algorithm. Theoretical justification is given in Section 3. The numerical and simulation studies are included in Section 4. Section 5 includes an application of the proposed method to a real data set and compares/combines it with other sketch algorithms. A brief discussion is provided in Section 6. All the technical proofs and additional numerical results and details are included in the Appendix.

## 2 REGULARIZATION IN REPRODUCING KERNEL HILBERT SPACES

Consider a continuous response $y_i$ and a $p$-dimensional covariate vector $X_i$, modeled as

$$y_i = f_0(X_i) + \epsilon_i, \quad i = 1, \ldots, N,$$

where $\epsilon_i$ are i.i.d. errors with mean zero and variance $\sigma^2$. We assume $f_0$ belongs to a reproducing kernel Hilbert space (RKHS) $\mathscr{H}_K$ induced by a symmetric, positive-definite kernel $K : \mathscr{X} \times \mathscr{X} \to \mathbb{R}$. If $\mu$ is a finite measure on $\mathscr{X}$ and $\int K(x,x)d\mu(x) < \infty$, applying Mercer's theorem (Mercer, 1909), one can write $K(x,y) = \sum_{j=1}^{\infty} \lambda_j \psi_j(x)\psi_j(y)$, where $\{\psi_j\}$ form an orthonormal basis in $L^2(\mu)$, $\lambda_j > 0$, and $\sum_{j=1}^{\infty} \lambda_j < \infty$. The RKHS is $\mathscr{H}_K = \left\{ f(x) = \sum_{j=1}^{\infty} c_j \psi_j(x) : \sum_{j=1}^{\infty} c_j^2/\lambda_j < \infty \right\}$, with norm $\|f\|_{\mathscr{H}_K}^2 = \sum_{j=1}^{\infty} c_j^2/\lambda_j$. To estimate $f_0$, we solve

$$\hat{f}_{N,\lambda^\star} = \arg \min_{f \in \mathscr{H}_K} \frac{1}{N} \sum_{i=1}^{N} \{y_i - f(X_i)\}^2 + \lambda^\star \|f\|_{\mathscr{H}_K}^2,$$

where $\lambda^\star > 0$ is a tuning parameter. By applying representer theorem (Wahba, 1990; Kimeldorf & Wahba, 1971), the solution has the finite expansion $\hat{f}_{N,\lambda^\star}(x) = \sum_{l=1}^{N} a_l K(x, X_l)$, with coefficients $\boldsymbol{a} = [a_1, \cdots, a_N]^\top$. Plugging this into the above objective for estimating $f_0$, it yields an objective function for $\boldsymbol{a}$, which is $J(\boldsymbol{a}) = \frac{1}{N}\|\boldsymbol{y} - \mathbf{K}\boldsymbol{a}\|^2 + \lambda^\star \boldsymbol{a}^\top \mathbf{K}\boldsymbol{a}$, where $\mathbf{K} \in \mathbb{R}^{N \times N}$ has entries $K(X_i, X_j)$. The solution $\hat{\boldsymbol{a}}$ that minimizes $J(\boldsymbol{a})$ is $\hat{\boldsymbol{a}} = N^{-1}\lambda^{\star-1}(\mathbf{I} + N^{-1}\lambda^{\star-1}\mathbf{K})^{-1}\boldsymbol{y}$. Thus, for any $x$,

$$\hat{f}_{N,\lambda^\star}(x) = N^{-1}\lambda^{\star-1}[K(x,X_1), \cdots, K(x,X_N)](\mathbf{I} + N^{-1}\lambda^{\star-1}\mathbf{K})^{-1}\boldsymbol{y}.$$

This requires inverting an $N \times N$ matrix, which is computationally demanding for large $N$. Moreover, performance depends on selecting $\lambda^\star$, adding to the computational cost.

### 2.1 PROPOSED METHOD FOR REGULARIZATION IN AN RKHS

Given data pairs $(X_1, y_1), \ldots, (X_N, y_N)$ for a large $N$, the goal is to estimate a nonparametric function $f_0(x)$ for a given $x$. Because $N$ is very large, a direct application of kernel machine method is time-consuming even not possible due to the inverse of an $N \times N$ matrix. Given a limited computational resource, we consider a sub-data selection approach by selecting a subsample $(X_{k_1}, y_{k_1}), \ldots, (X_{k_n}, y_{k_n})$ with size $n$ from the original sample with size $N$ while maximizing the statistical efficiency of the resulting kernel machine estimator $\hat{f}_{n,\tilde{\lambda}^\star}(x)$, where $\tilde{\lambda}^\star$ denotes the tuning parameter for the proposed method.

Our proposed procedure makes use of the smoothness property of the nonparametric functions by approximating the functional values of $f_0(x)$ by a set of representative data points. To find the representative points, we firstly apply a clustering approach to cluster $N$ data points into $L$ representative clusters $\{\mathcal{C}_1, \ldots, \mathcal{C}_L\}$, and then re-sampling with optimal weights $\{\omega_{x,1,C}, \ldots, \omega_{x,L,C}\}$ from these $L$ clusters to obtain these representative data points. Euclidean k-means is a common choice for clustering, it is appropriate for kernels that preserve local Euclidean structure (such as an RBF kernel). For RKHS spaces introduced by more general kernels, clustering algorithms such as kernel K-means or K-medians (Wang et al., 2019b) that preserve the geometry structure of the RKHS space are better choices.

---

**Algorithm 1** Proposed Weighted Resampling RKHS Estimator

---

**Require:** Data $\{(X_i, y_i)\}_{i=1}^N$, subsample size $n$, number of clusters $L$, number of resamples $B$ and kernel $K$

**Ensure:** $\hat{f}_{n,\tilde{\lambda}^\star}(x)$

1: **Clustering.** Given $X_1, \ldots, X_N$ and subsample size $n$, apply a clustering method such as K-means (or random projected K-means or kernel K-means or K-medians) to partition the dataset into $L$ clusters $\mathcal{C}_1, \ldots, \mathcal{C}_L$. Let the cluster centers be $\{C_1, \ldots, C_L\}$.

2: **Tuning $\tilde{\lambda}^\star$.** Apply cross-validation or BIC method to find the tuning parameter $\tilde{\lambda}^\star$ for the RKHS regression using the representative points $\{C_1, \ldots, C_L\}$.

3: **Assigning weights.** Using $\tilde{\lambda}^\star$, compute optimal cluster weights $\omega_{x,1,C}, \ldots, \omega_{x,L,C}$ as defined in (2), and compute weights $\omega_i(x_0)$ for each data points defined in Section 3.2.

4: **for** $b = 1$ to $B$ **do**

5:     **Resampling with weights.** Using the weights $\omega_1(x), \ldots, \omega_N(x)$ to sample $n$ points from $\{X_1, \ldots, X_N\}$ to obtain $\{X_1^\star, \ldots, X_n^\star\}$ and corresponding outcomes $\boldsymbol{y}^\star = [y_1^\star, \ldots, y_n^\star]^\top$.

6:     **Compute RKHS estimator.** Using the selected sample and $\tilde{\lambda}^\star$ to compute $\hat{f}_{n,\tilde{\lambda}^\star}^{(b)}(x)$ which is defined similarly to $\hat{f}_{N,\lambda^*}(x)$ except that it uses the $b$-th sampling data.

7: **end for**

8: **Aggregate:** The final estimate of $f_0(x)$ is $\hat{f}_{n,\tilde{\lambda}^\star}(x) = \dfrac{1}{B} \sum_{b=1}^B \hat{f}_{n,\tilde{\lambda}^\star}^{(b)}(x)$.

---

The optimal weights $\{\omega_{x,1,C}, \ldots, \omega_{x,L,C}\}$ are chosen to maximize the statistical efficiency of the proposed estimator. Let $\{\mathcal{C}_1, \ldots, \mathcal{C}_L\}$ denote the clusters, where $\mathcal{C}_l$ contains $N_l$ observations with $\sum_{l=1}^L N_l = N$. We denote the center of cluster $\mathcal{C}_l$ by $C_l$. Suppose we select a subdata set with size $n$, among them $n_l$ data points are from the $l$-th cluster so that $n_l = n\omega_{x,l,C}$ and $\sum_{l=1}^L \omega_{x,l,C} = 1$. For all the data selected from the cluster $\mathcal{C}_l$, we approximate them using their representative data centers $C_l$. With a slightly abuse of notations, we rearrange the data $\{y_1, \ldots, y_N\}$ by clusters and denote $y_{lj}$ the $j$-th data points selected from the $l$-th cluster. Then, the corresponding model for the $l$-th cluster mean $\bar{y}_l = \sum_{j=1}^{n_l} y_{lj}/n_l$ is

$$\bar{y}_l \approx f_0(C_l) + \bar{\epsilon}_l,$$

where $\bar{\epsilon}_l = \sum_{j=1}^{n_l} \epsilon_{lj}/n_l$ has mean zero and variance $\sigma^2/(n\omega_{x,l,C})$. We then consider the following RKHS nonparametric estimation of $f_0(x)$ given the above model:

$$\hat{f}_{n,\tilde{\lambda}^\star}(x) = L^{-1}\tilde{\lambda}^{\star-1}\big[K(x,C_1), \cdots, K(x,C_L)\big]\big(\mathbf{I} + L^{-1}\tilde{\lambda}^{\star-1}\mathbf{K}_c\big)^{-1}\bar{\boldsymbol{y}},$$

where $\mathbf{K}_c$ is an $L \times L$ matrix with $(i,j)$-th element $K(C_i, C_j)$ and $\bar{\boldsymbol{y}} = [\bar{y}_1, \cdots, \bar{y}_L]^\top$. Conditional on the centers $\{C_1, \ldots, C_L\}$, the variance of the estimate $\hat{f}_{n,\tilde{\lambda}^\star}(x)$ is therefore

$$\text{Var}\{\hat{f}_{n,\tilde{\lambda}^\star}(x)\} = \sigma^2 L^{-3}\tilde{\lambda}^{\star-2}\mathbf{K}_x(C)^\top\big(\mathbf{I} + L^{-1}\tilde{\lambda}^{\star-1}\mathbf{K}_c\big)^{-1}\mathbf{D}_\omega\big(\mathbf{I} + L^{-1}\tilde{\lambda}^{\star-1}\mathbf{K}_c\big)^{-1}\mathbf{K}_x(C),$$

where $\mathbf{K}_x(C) = \big[K(x,C_1), \cdots, K(x,C_L)\big]^\top$, $\mathbf{D}_\omega = \text{diag}\{1/\omega_{x,1,C}, \cdots, 1/\omega_{x,L,C}\}$. Because $\mathbb{E}(\bar{\mathbf{y}}) = [f_0(C_1), \cdots, f_0(C_L)]^\top$, the conditional bias of the estimator $\hat{f}_{n,\tilde{\lambda}^\star}(x)$ is independent of the choices of $\omega_{x,i,C}$'s. Therefore, to minimize the conditional MSE of $\hat{f}_{n,\tilde{\lambda}^\star}(x)$, we could choose $\omega_{x,l,C}$'s that minimize the variance of $\hat{f}_{n,\tilde{\lambda}^\star}(x)$:

$$\omega_{x,l,C} = \arg\min_{\omega_{x,l,C} \geq 0, \sum \omega_{x,l,C} = 1} \text{Var}\{\hat{f}_{n,\tilde{\lambda}^\star}(x)\}. \tag{1}$$

It is not difficult to check that $\text{Var}\{\hat{f}_{n,\tilde{\lambda}^\star}(x)\} = L^{-3}\sigma^2\tilde{\lambda}^{\star-2}\sum_{l=1}^L K_{xl}^2/\omega_{x,l,C}$. where $K_{xl}$ is the $l$-th element of the vector $\boldsymbol{K}_x$ which is defined by

$$\boldsymbol{K}_x = \big(\boldsymbol{I} + L^{-1}\tilde{\lambda}^{\star-1}\boldsymbol{K}_c\big)^{-1}\begin{bmatrix} K(x,C_1) \\ \vdots \\ K(x,C_L) \end{bmatrix} = \begin{bmatrix} K_{x1} \\ \vdots \\ K_{xL} \end{bmatrix}.$$

Therefore, the optimal weights $\omega_{x,l,C}$'s that minimize the variance is

$$\omega_{x,l,C} = |K_{xl}| / \sum_{l=1}^{L} |K_{xl}|. \tag{2}$$

Because the above optimal weights depend on the tuning parameter $\tilde{\lambda}^\star$, an initial tuning parameter is needed to determine the optimal weights. We have compared an iterative algorithm with a one-step iterative algorithm, and we found that their performance were similar. To save computational time, we adapted the one-step iterative procedure in Algorithm 1.

For the tuning procedure in Algorithm 1, it requires $\hat{f}_{n,\tilde{\lambda}^\star}(x)$ which is an RKHS estimator of $f_0(x)$ based on the centers selected. However, the estimator depends on the tuning parameter $\tilde{\lambda}^\star$, embedded in the inverse operation, which needed to computed for every possible candidates $\lambda$. This leads to a big computational burden. To mitigate computational burden for selecting the tuning parameter $\tilde{\lambda}^\star$, we decompose the kernel matrix for the selected data with size $n$ into $Q\Lambda Q^\top$, where $Q$ is an orthogonal matrix that is independent of $\tilde{\lambda}^\star$, and $\Lambda = \text{diag}\{\theta_1, \cdots, \theta_n\}$ is an $n \times n$ diagonal matrix of eigenvalues. Using the notations defined in Algorithm 1, we can write $\hat{f}_{n,\tilde{\lambda}^\star}(x) = [K(x, X_1^\star), \cdots, K(x, X_n^\star)]^\top Q\Lambda_{\tilde{\lambda}^\star}^{-1}Q^\top y^\star$, where $y^\star = [y_1^\star, \cdots, y_n^\star]^\top$ is an $n \times 1$ vector and $\Lambda_{\tilde{\lambda}^\star} = \text{diag}\{\tilde{\lambda}^\star + \theta_1, \cdots, \tilde{\lambda}^\star + \theta_n\}$. Then, one only needs to compute the inverse of a diagonal matrix $\Lambda_{\tilde{\lambda}^\star}$ for different tuning parameters.

## 3 THEORETICAL JUSTIFICATION

### 3.1 ASYMPTOTIC RATE AND TUNING WITH FULL DATA

*Theorem* 1. Assume $X$ has a probability density function $\pi(x)$ and $\mathbb{E}\{\psi_j^4(X)\} < \infty$. If $\lambda_j$ decays at the order of $j^{-k}$ for some $k > 1$ where $\lambda_j$ is the $j$-th largest eigenvalue of the positive definite $K(\cdot, \cdot)$ and $k$ is the decaying rate. For any $f \in \mathscr{H}_K$ with expansion $f(x) = \sum_{j=1}^{\infty} c_j \psi_j(x)$ and the coefficients satisfy $c_j^2 \lambda_j^{-1} \asymp j^{-a}(a > 1)$, the optimal rate of tuning parameter is: $\lambda^\star \asymp N^{-\frac{k}{k+1+(a-1)}}$, and the corresponding asymptotic integrated mean squared error (IMSE) is $\|\hat{f}_{N,\lambda^\star} - f_0\|^2 = O_p\{N^{-\frac{(a-1)+k}{(a-1)+k+1}}\}$, where $\|g\|$ is the $L_2$ norm of a function $g$.

The assumptions on eigenvalues and eigenfunctions of kernel are common in the literature (e.g.,Yuan & Cai (2010); Zhang et al. (2015)). The details of the proof of Theorem 1 are given in the Appendix. Comparing with the results in the existing literature (e.g., Yuan & Cai (2010)), the results in Theorem 1 give a more specific rate of convergence for the functions $f \in \mathscr{H}_K$ with a specific form specified in Theorem 1. If we consider all the possible functions in $\mathscr{H}_K$, the rate of convergence is given by (letting $a \to 1$) $\|\hat{f}_{N,\lambda^\star} - f_0\|^2 = O_p(N^{-\frac{k}{k+1}})$. For functions in a univariate Sobolev space of order $m$, the eigenvalue of the corresponding reproducing kernel is at the order of $\lambda_j \asymp j^{-2m}$ Yuan & Cai (2010), then the rate of the convergence is given by $\|\hat{f}_{N,\lambda^\star} - f_0\|^2 = O_p(N^{-\frac{2m}{2m+1}})$. This rate is known to be optimal in the literature (e.g. Zhang et al. (2015)).

Theorem 1 has multi-purposes. First, the rate of convergence results of Theorem 1 is a basis to define an efficiency index, which is designed to measure the trade-off between computational and statistical efficiency. This would facilitate the comparison among different methods. The details are given in Section 4. Second, Theorem 1 provides a general guidance about the order of tuning parameter choices subject to a constant, which can be further selected by the BIC or cross-validation.

### 3.2 ASYMPTOTIC RATE AND TUNING WITH RESAMPLING UNDER PROPOSED METHOD

For a given $x_0$, the proposed resampling weights for a sample $X_i \in \mathcal{C}_l$ where $\mathcal{C}_l$ is the $l$-th cluster whose center is $C_l$ with a cluster size $N_l$, the weight for $X_i$ to be selected is: $\omega_i(x_0) = N_l^{-1}\omega_{x_0,l,C}$, where $\omega_{x_0,l,C}$ was defined in (2). For each $x_0$ and the $j$-th eigenfunctions, define $\omega_{x_0,j} = \mathbb{E}\{\sum_{i=1}^{N} \omega_i(x_0) \psi_j^2(X_i)\}$. Denote the indices of the sampled data points with size $n$ for a given $x_0$ by $\mathscr{S}(x_0) = \{i_1^*, \ldots, i_n^*\} \subset \{1, \ldots, N\}$, and the sampled data points are $\{(X_i, y_i) : i \in \mathscr{S}(x_0)\}$.

The following Lemma studies the weight function $\omega_{x_0,j}$. The detailed proof of Lemma 1 can be found in the Appendix.

**Lemma 1.** *Let $\omega_{x_0,l,C}$ be the cluster-level weights defined by (2), and assign each sample $X_i \in \mathcal{C}_l$ the resampling weights $\omega_i(x_0) = N_l^{-1} \omega_{x_0,l,C}$. Define $\omega_{x_0,j} = \mathbb{E}\left\{ \sum_{i=1}^N \omega_i(x_0)\psi_j^2(X_i) \right\}$. If $\lambda_j \asymp j^{-k}$ and $\int K^2(x,y)d\pi(y) < \infty$, then $\omega_{x_0,j} \le \lambda_j^{-2} \asymp j^{2k}$.*

The proposed estimator of $f(x_0)$ is $\hat{f}_{n,\tilde{\lambda}^\star,x_0}(x_0)$, which is given by:

$$\hat{f}_{n,\tilde{\lambda}^\star,x_0} = \arg\min_{f \in \mathscr{H}_K} \left[ \frac{1}{n} \sum_{i \in \mathscr{S}(x_0)} \{y_i - f(X_i)\}^2 + \tilde{\lambda}^\star \|f\|_{\mathscr{H}_K}^2 \right].$$

Define the functional $\hat{f}_{n,\tilde{\lambda}^\star}$ of the proposed estimator by collecting the estimators at all $x_0 \in \mathscr{X}$ together $\hat{f}_{n,\tilde{\lambda}^\star} = \left\{ \hat{f}_{n,\tilde{\lambda}^\star,x_0}(x_0) : x_0 \in \mathscr{X} \right\}$, where $\mathscr{X}$ is the space of the predictor/feature $x$.

*Theorem* 2. Assume $X$ has a probability density function $\pi(x)$ and $\mathbb{E}\{\psi_j^4(X)\} < \infty$. If $\lambda_j$ decays at the order of $j^{-k}$ for some $k > 1$ where $\lambda_j$ is the $j$-th largest eigenvalue of the positive definite $K(\cdot,\cdot)$ and $k$ is the decaying rate. Assume $\omega_{x_0,j} \asymp j^\beta$ for $0 \le \beta < k$. Then, for any function $f_0$ in RKHS, the optimal rate of the tuning parameter is: $\tilde{\lambda}^\star \asymp n^{-\frac{k-\beta}{k+1-2\beta}}$ if $\beta \le 1/2$ and $\tilde{\lambda}^\star \asymp n^{-1/2}$ if $\beta > 1/2$, and the corresponding asymptotic IMSE of the estimator $\hat{f}_{n,\tilde{\lambda}^\star}$ is

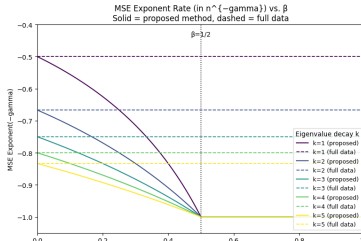

$$\left\| \hat{f}_{n,\tilde{\lambda}^\star} - f_0 \right\|^2 = \begin{cases} O_p\left(n^{-\frac{k}{k+1-2\beta}}\right) & \text{if } \beta \in [0,1/2), \\ O_p\left(n^{-1}\log(n)\right) & \text{if } \beta = 1/2, \\ O_p\left(n^{-1}\right) & \text{if } \beta > 1/2. \end{cases}$$

Figure 1: Asymptotic MSE for the proposed method for different values of $\beta$ and $k$. Solid curves show the proposed method (using the optimal $\tilde{\lambda}^\star$), and dashed curves show the rate the full-data estimator for comparison.

The assumption $\omega_{x_0,j} \asymp j^\beta$ in Theorem 2 is based on the result in Lemma 1. Detailed proof of Theorem 2 is given in the Appendix. Based on the results in Theorem 2, the rate of convergence of the proposed estimator is $\left\| \hat{f}_{n,\tilde{\lambda}^\star} - f_0 \right\|^2 = O_p\left(n^{-\frac{k}{k+1-2\beta}}\right)$ or $O_p(n^{-1})$ for any functions in the RKHS $\mathscr{H}_K$. Because $\beta \ge 0$, the rate of the proposed estimator is faster than the rate of the RKHS estimator sampled by the SRS with the same subsample size, which is given by $n^{-\frac{k}{k+1}}$. The proposed method leads to more substantial improvement when $\beta \ge 1/2$, where the optimum reaches the parametric rate $n^{-1}$ with sample size $n$. The improvement in the estimator rate is due to informative sampling, where the sampling weights carry the information of the entire data set. An illustration of the rate improvement is provided in Figure 1.

## 4 SIMULATION STUDY

### 4.1 PROPOSED VS. SIMPLE RANDOM SAMPLE VS. FULL DATA

Starting with the simplest case, we evaluate the numerical performance of our proposed method in estimating the unknown function $f_0(x) = 5\cos x$, and compare it with that based on the full data and data sampled using a simple random sampling (SRS) strategy. The full sample size is denoted by $N$ and the subsample size is denoted by $n$. The random errors $\epsilon_1, \ldots, \epsilon_n \sim N(0, 0.5)$ i.i.d.. The kernel function $K(\cdot,\cdot)$ was Gaussian. Results are based on 100 replications, and $B = 1$ was used for the proposed method. The simulation study was implemented in an R environment.

To evaluate each method, we compute the integrated mean squared error (IMSE) and record the computational time. To compare methods $A$ and $R$, we use relative efficiency (RE):

$$\text{RE} = \frac{\text{Efficiency}_A}{\text{Efficiency}_R} = \frac{(\text{IMSE}_A)^{-\alpha}(\text{Time}_A)^{-\beta}}{(\text{IMSE}_R)^{-\alpha}(\text{Time}_R)^{-\beta}},$$

Table 1: IMSE, average timing (hr:min'sec") and relative efficiency (RE) for $f_0(x) = 5\cos(x)$.

| | | Full Data | | Proposed | | | SRS | | |
|---|---|---|---|---|---|---|---|---|---|
| $N$ | $n$ | IMSE | Time | IMSE | Time | RE | IMSE | Time | RE |
| 1000 | 100 | 0.0034 | 0:0'2 | 0.0074 | 0:0'0 | 2.20 | 0.0297 | 0:0'0 | 0.68 |
| 1000 | 200 | 0.0034 | 0:0'2 | 0.0046 | 0:0'0 | 2.66 | 0.0152 | 0:0'0 | 0.89 |
| 5000 | 500 | 0.0008 | 0:3'35 | 0.0017 | 0:0'0 | 3.26 | 0.0067 | 0:0'0 | 0.85 |
| 5000 | 1000 | 0.0008 | 0:3'35 | 0.0011 | 0:0'3 | 2.72 | 0.0036 | 0:0'2 | 0.76 |
| 10000 | 1000 | 0.0005 | 0:22'16 | 0.0009 | 0:0'3 | 3.49 | 0.0036 | 0:0'2 | 0.76 |
| 10000 | 2000 | 0.0005 | 0:22'16 | 0.0006 | 0:0'26 | 2.98 | 0.0020 | 0:0'14 | 0.81 |

where $R$ is the full-data estimator and we choose $\alpha = 5/4$, $\beta = 1/3$ so that the efficiency of the full data method is (asymptotically) a constant with respect to $N$ because Theorem 1 suggests that the full data approach has IMSE $\asymp N^{-4/5}$ for functions in the Sobolev space with order $m = 2$ and computation time $\asymp N^3$ (using direct computation), and the RE of the method based on the SRS is approximately 1. RE $< 1$ indicates method A performed worse than the full data method; RE $> 1$ indicates method A performed better than the full data method.

## 4.2 SIMULATION STUDY: FALKON VS. PROPOSED VS. NYSTRÖM

We compare the proposed K-means–based resampling method with two fast kernel ridge regression baselines: FALKON methods including FALKON17 (Rudi et al., 2017) and FALKON20 (Meanti et al., 2020), and Nyström KRR (Williams & Seeger, 2001). We generate 20-dimensional predictors with five relevant variables and evaluate training sizes $N \in \{2000, 5000, 10000\}$ under matched computational budgets using subset sizes $n \in \{100, 500, 1000\}$. Across 100 replications, we report test MSE and total runtime (including tuning). Detailed experimental setup, tuning grids, and implementation notes are provided in Appendix E.1.

Table 2: Simulation results: mean squared error (MSE) and total time (seconds) averaged over 100 replications, $n$ is the subset size ($M$ for FALKON/Nyström and $n$ for Proposed), $B = 3$ for Proposed. Time includes tuning and final training.

| | | MSE | | | | Time | | |
|---|---|---|---|---|---|---|---|---|
| $N$ | $n$ | FALKON17 | FALKON20 | Proposed | Nyström | FALKON17 | Proposed | Nyström |
| 2000 | 100 | 6.7196 | 6.7001 | **5.7745** | 6.5325 | 0.0615 | 0.1241 | 0.0166 |
| 2000 | 500 | 4.1357 | 4.1636 | **2.2412** | 2.5114 | 0.3186 | 0.7966 | 0.5946 |
| 2000 | 1000 | 3.6189 | 3.6094 | **1.7058** | 2.0939 | 0.8643 | 3.2982 | 1.2725 |
| 5000 | 100 | 6.4308 | 6.4577 | **5.7397** | 6.2693 | 0.1898 | 0.1805 | 0.0234 |
| 5000 | 500 | 2.9266 | 2.9216 | **1.8134** | 2.3298 | 0.6127 | 1.6280 | 0.6692 |
| 5000 | 1000 | 2.4558 | 2.4484 | **1.2729** | 1.9035 | 1.5446 | 7.1565 | 1.6389 |
| 10000 | 100 | 6.3071 | 6.3295 | **5.7606** | 6.2087 | 0.2486 | 0.3953 | 0.0365 |
| 10000 | 500 | 2.3723 | 2.3875 | **1.6411** | 2.0700 | 1.0347 | 3.0601 | 0.7498 |
| 10000 | 1000 | 2.0695 | 2.0586 | **1.0393** | 1.7595 | 2.3338 | 13.2765 | 1.9417 |

Table 2 shows that the proposed method attains the lowest MSE for all the sub-data considered in the simulation, reflecting the benefit of informative resampling. FALKON remains the fastest, especially at small $n$, while Nyström is competitive but trails the proposed estimator in accuracy under the similar computational budget.

## 5 REAL–DATA STUDY: YEARPREDICTIONMSD

### 5.1 METHOD COMPARISON: PROPOSED VS. FALKON VS. NYSTRÖM

We evaluate the proposed estimator on the YEARPREDICTIONMSD dataset (90 features, continuous response) and compare it with Nyström KRR and FALKON under matched computational budgets.

For each experiment we draw $N \in \{2000, 5000, 10000, 20000\}$ for training and use a fixed test set of size 1000, with all randomization seeded for reproducibility. We also apply a simple filter step by selecting the top-$p$ features where $p \in \{30, 60, 90\}$. All methods use RBF kernels unless otherwise specified, with subset/landmark budget $n = M$ for Nyström and FALKON, and $n$ as subsample size for the proposed method. Full implementation details, kernel settings, and runtime protocol are provided in Appendix E.2.

Table 3: YearPredictionMSD data using kernel learning with 1,000 testing data points. Proposed method uses random-projection-$k$-means centers, $B$=3, fixed $\lambda^*$=1, with three kernels: Prop.G = Gaussian RBF, Prop.M = Matérn-3/2, Prop.L = Laplace. Times for Proposed are averaged across these three kernels and shown as total with centers-time in parentheses. FALKON and Nyström use Gaussian RBF ($\sigma$=6). FAL17 and FAL20 refer to the implementations of Rudi et al. (2017) and Meanti et al. (2020), respectively.

| | | | MSE | | | | | | Time (s) | | |
|---|---|---|---|---|---|---|---|---|---|---|---|
| $N$ | $n$ | $p$ | Prop.G | Prop.M | Prop.L | FAL17 | FAL20 | Nyström | Proposed | FALKON | Nyström |
| 2000 | 500 | 30 | 100.67 | 97.76 | **96.48** | 122.42 | 110.40 | 124.54 | 1.32 (0.16) | 0.12 | 0.11 |
| | | 60 | 96.26 | **93.87** | 94.14 | 105.75 | 97.93 | 109.25 | 1.22 (0.08) | 0.06 | 0.05 |
| | | 90 | 96.96 | 95.36 | 95.34 | 96.65 | **95.06** | 98.38 | 1.12 (0.07) | 0.06 | 0.06 |
| 5000 | 1000 | 30 | 94.55 | 92.80 | **91.44** | 108.61 | 103.29 | 116.52 | 4.10 (0.17) | 0.10 | 0.14 |
| | | 60 | 95.37 | **92.56** | 92.98 | 105.11 | 106.85 | 105.20 | 3.95 (0.18) | 0.09 | 0.16 |
| | | 90 | **87.35** | 88.98 | 89.07 | 91.57 | 93.15 | 91.59 | 3.25 (0.16) | 0.13 | 0.16 |
| 5000 | 2000 | 30 | 81.39 | 81.22 | **80.51** | 113.93 | 105.82 | 154.15 | 15.05 (0.20) | 0.26 | 0.61 |
| | | 60 | 78.45 | 78.79 | **78.42** | 95.62 | 103.94 | 99.60 | 14.19 (0.17) | 0.26 | 0.63 |
| | | 90 | **76.38** | 76.62 | 77.18 | 86.14 | 101.82 | 88.08 | 14.13 (0.16) | 0.25 | 0.68 |
| 10000 | 1000 | 30 | 92.90 | 92.33 | **92.22** | 97.37 | 100.86 | 101.36 | 4.58 (0.41) | 0.13 | 0.27 |
| | | 60 | 91.16 | **89.99** | 91.14 | 97.00 | 98.03 | 93.86 | 4.50 (0.45) | 0.13 | 0.18 |
| | | 90 | **88.38** | 88.53 | 91.41 | 90.63 | 88.72 | 91.26 | 4.56 (0.44) | 0.10 | 0.18 |
| 10000 | 2000 | 30 | 96.52 | 95.73 | **95.16** | 110.34 | 101.30 | 115.29 | 15.83 (0.48) | 0.27 | 0.71 |
| | | 60 | **88.77** | 89.86 | 90.75 | 94.10 | 99.89 | 95.96 | 15.48 (0.42) | 0.31 | 0.72 |
| | | 90 | 87.31 | **86.63** | 88.73 | 87.63 | 92.06 | 87.38 | 15.47 (0.50) | 0.26 | 0.73 |
| 20000 | 1000 | 30 | 97.05 | 96.24 | 95.75 | 97.44 | **95.37** | 99.82 | 9.84 (1.45) | 0.22 | 0.23 |
| | | 60 | 96.93 | 96.82 | **95.11** | 96.95 | 95.19 | 98.38 | 10.38 (1.40) | 0.15 | 0.25 |
| | | 90 | 94.05 | 94.42 | 95.14 | 95.02 | **89.42** | 97.59 | 10.13 (1.38) | 0.15 | 0.25 |
| 20000 | 2000 | 30 | 98.24 | 98.25 | 98.34 | 105.48 | **94.60** | 110.94 | 20.56 (1.42) | 0.33 | 0.92 |
| | | 60 | 98.50 | 99.32 | 97.92 | 101.90 | **94.21** | 103.28 | 20.38 (1.27) | 0.33 | 1.00 |
| | | 90 | 94.17 | 92.97 | 94.23 | 98.11 | **87.58** | 93.95 | 19.79 (1.34) | 0.33 | 0.93 |

Table 3 summarizes the test MSEs and runtimes. The proposed method is consistently competitive with (and often outperforms) FALKON and Nyström in accuracy, with longer but manageable computation time. This is because our proposed method is pointwise and for each test data point, a sub-data is selected. The Laplace and Matérn variants provide additional robustness, with Gaussian RBF performing strongest in some settings. Although the proposed method is designed for sub-data selection for pointwise prediction, the reported run times show that the proposed approach scales well for $N$=20k with $n$=2000, where total time cost remains at the order of 20 seconds. An investigation of the impact of clustering methods on the proposed method is given in the Appendix.

## 5.2 Pointwise Prediction

Since our method is designed for pointwise prediction, we conduct an experiment to demonstrate how it can be used to improve the computational efficiency of FALKON while maintaining statistical accuracy. As discussed in Section 3.2, combining our sub-data selection with FALKON allows pointwise prediction to be performed on a much smaller but informative subset of the data, reducing training cost without sacrificing estimation quality. In this experiment, FALKON is trained once on the full training set of size $N$ using $M$ randomly chosen landmarks (Gaussian kernel with $\sigma = 6$, $\lambda^* = 10^{-6}$, 20 iterations), and then used to predict all 200 test points. In contrast, Prop+FALKON

begins with $k$-means clustering on all $N$ points, computes sampling weights for each test point $x_0$, draws a pointwise subdata of size $n \asymp 3N^{0.8}$, and fits FALKON with the same $M$ landmarks on this subset to make the prediction at $x_0$.

Table 4: Numerical comparison of pointwise prediction on `YearPredictionMSD` using full-data FALKON and the proposed+FALKON (Prop+F) method across different training sizes $N$, subdata sizes $n$, and landmark counts $M$. Timing columns give the average computation per test point. Clust is the $k$-means clustering time for Prop+F, Train is the training time for Prop+F, and Sel is the sampling/weighting time for Prop+F.

| Sample sizes | | | MSE | | RMSE | | Total Time (s) | | Time dist. (Prop+F) | | |
|---|---|---|---|---|---|---|---|---|---|---|---|
| $N$ | $M$ | $n$ | FALKON | Prop+F | FALKON | Prop+F | FALKON | Prop+F | Clust | Train | Sel |
| 10k | 0.5k | 4.76k | 81.035 | 83.809 | 9.002 | 9.153 | 0.028 | 0.273 | 0.234 | 0.018 | 0.021 |
| 20k | 1k | 8.28k | 88.075 | 87.974 | 9.383 | 9.382 | 0.107 | 0.851 | 0.713 | 0.067 | 0.071 |
| 50k | 1k | 17.23k | 78.571 | 78.139 | 8.865 | 8.840 | 0.228 | 3.991 | 3.467 | 0.101 | 0.423 |
| 100k | 5k | 30.00k | 80.573 | 85.890 | 8.977 | 9.273 | 3.522 | 11.765 | 8.781 | 1.844 | 1.140 |
| 100k | 10k | 30.00k | 85.107 | 83.930 | 9.224 | 9.160 | 56.999 | 17.457 | 9.062 | 7.261 | 1.134 |
| 200k | 5k | 52.23k | 82.603 | 77.079 | 9.090 | 8.778 | 47.972 | 26.735 | 21.687 | 2.162 | 2.886 |
| 200k | 10k | 52.23k | 75.708 | 78.909 | 8.697 | 8.888 | 96.188 | 33.800 | 22.693 | 8.319 | 2.788 |

The pointwise results in Table 4 demonstrate that the proposed selection strategy can substantially reduce the effective training size while preserving predictive accuracy. Across all training sizes, the subdata-based estimator achieves MSE and RMSE comparable to (and occasionally better than) full FALKON, despite operating on significantly smaller subsets. The reduction in training points also leads to computation reduction, particularly for large $N = 200k$ without compromising performance. As for computational complexity, we compare the theoretical time and memory costs in Table 5.

Table 5: Time and memory complexity of kernel methods. $N$ = full training size, $M$ = number of landmarks, $n \asymp N^\gamma$ subdata size selected by the proposed method, where $\gamma = (k + 1 - 2\beta)/(k + 1)$ if $\beta \in [0, 1/2)$ and $\gamma = k/(k + 1)$ if $\beta > 1/2$, $k$ is defined in Theorem 2. $K$ = number of clusters, and $t$ = number of iterations. Details in E.2 of the Appendix.

| Method | Time Complexity | Memory Complexity |
|---|---|---|
| FALKON (Rudi et al. (2017); $M \asymp \sqrt{N}$) | $O(N^{3/2})$ | $O(N)$ |
| FALKON (Meanti et al. (2020)) | $O(NM) + O(M^2)$ | $O(NM + M^2)$ |
| Proposed + FALKON | $O(N^\gamma M + N^\gamma Kt)$ | $O(N^\gamma M + M^2 + K)$ |

## 6 DISCUSSION

A growing body of recent work has demonstrated theoretical equivalences and connections between deep neural networks (DNNs) and kernel learning methods (e.g., Jacot et al. (2018); Zhu et al. (2022); Zhang et al. (2024)). As highlighted in Belkin et al. (2018), developing a deeper understanding of more tractable kernel methods is an important step toward building a solid theoretical foundation for DNNs. This paper contributes to the understanding of kernel learning for large-scale data sets, both theoretically and empirically. We address the computational challenges of kernel learning by selecting an informative, prediction-oriented subset of the data, allowing kernel machines to operate on a much smaller effective sample size. Unlike sketching methods that approximate the kernel matrix, our approach selects informative samples by assigning sampling weights that minimize the prediction MSE.

A key finding is that the proposed method can improve the convergence rate by sampling from the full data in an informative manner. When integrated with FALKON, our method substantially reduces FALKON's computational cost on the full data while maintaining its predictive accuracy. Numerical results show that the proposed method achieves lower IMSE in estimating the unknown nonparametric function than SRS-based methods and offers IMSE comparable to the full-data estimator. It also delivers competitive out-of-sample predictive performance relative to existing sketching algorithms, such as FALKON and Nyström methods.

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

# A  PROOF OF THEOREM 1

In this proof, we derive the asymptotic mean squared error of the nonparametric estimator for full data. First, we recall that, the objective function to estimate $f_0(x)$ is given by:

$$\hat{f}_{N,\lambda^\star} = \arg\min_{f \in \mathscr{H}_K} \left\{ \frac{1}{N} \sum_{i=1}^{N} (y_i - f(X_i))^2 + \lambda^\star \|f\|_{\mathscr{H}_K}^2 \right\},$$

where the norm of any function $f$ in $\mathscr{H}_K$ is:

$$\|f\|_{\mathscr{H}_K}^2 := \sum_{j=1}^{\infty} \frac{c_j^2}{\lambda_j} = \sum_{j=1}^{\infty} \lambda_j \langle f, \psi_j \rangle_{\mathscr{H}_K}^2 < \infty.$$

So the objective function is equivalent to:

$$\hat{f}_{N,\lambda^\star} = \arg\min_{f \in \mathscr{H}_K} \left\{ \frac{1}{N} \sum_{i=1}^{N} (y_i - f(X_i))^2 + \lambda^\star \sum_{j=1}^{\infty} \frac{c_j^2}{\lambda_j} \right\}.$$

To investigate the asymptotic mean squared error, we will decompose the MSE of the estimator into deterministic error and stochastic error. More specifically, the estimation error between $\hat{f}_{N,\lambda^\star}$ and the true function $f_0$ can be decomposed as

$$\hat{f}_{N,\lambda^\star} - f_0 = (\hat{f}_{N,\lambda^\star} - \bar{f}_{\infty,\lambda^\star}) + (\bar{f}_{\infty,\lambda^\star} - f_0)$$

where we refer $\hat{f}_{N,\lambda^\star} - \bar{f}_{\infty,\lambda^\star}$ as stochastic error and $\bar{f}_{\infty,\lambda^\star} - f_0$ as deterministic error. Here $\bar{f}_{\infty,\lambda^\star}$ is the solution of the following objective function:

$$\bar{f}_{\infty,\lambda^\star} = \arg\min_{f \in \mathscr{H}_K} \{ l_\infty(f) + \lambda^\star \|f\|_{\mathscr{H}_K}^2 \}$$

where the loss function $l_\infty(f)$ is the limit of $l_N(f) = \frac{1}{N} \sum_{i=1}^{N} (y_i - f(X_i))^2$. That is

$$l_\infty(f) := \mathbb{E}(l_N(f)) = \mathbb{E}\left\{ y - f(X) \right\}^2$$
$$= \sigma^2 + \mathbb{E}\{f(X) - f_0(X)\}^2.$$

Define the functional $\hat{f}_{N,\lambda^\star}$ of the population estimator by collecting the estimators at all $x_0 \in \mathcal{X}$ together $\hat{f}_{N,\lambda^\star} = \left\{ \hat{f}_{N,\lambda^\star}(x_0) : x_0 \in \mathcal{X} \right\}$, where $\mathcal{X}$ is the space of the predictor/feature $x$. Define the norm:

$$\|\hat{f}_{N,\lambda^\star} - f\|^2 = \int < \hat{f}_{N,\lambda^\star} - f, K(x_0, \cdot) >_{\mathscr{H}_K}^2 \pi(x_0) dx_0$$

$$= \int < \hat{f}_{N,\lambda^\star} - f, \sum_{j=1}^{\infty} \lambda_j \psi_j(\cdot) \psi_j(x_0) >_{\mathscr{H}_K}^2 \pi(x_0) dx_0$$

$$= \int (\sum_{j=1}^{\infty} \lambda_j < \hat{f}_{N,\lambda^\star} - f, \psi_j(\cdot) >_{\mathscr{H}_K} \psi_j(x_0))^2 \pi(x_0) dx_0$$

$$= \sum_{j=1}^{\infty} \lambda_j^2 < \hat{f}_{N,\lambda^\star} - f, \psi_j(\cdot) >_{\mathscr{H}_K}^2$$

$$= \sum_{j=1}^{\infty} \lambda_j^2 < \sum_{l=1}^{\infty} (\hat{c}_l - c_l) \psi_l, \psi_j >_{\mathscr{H}_K}^2$$

$$= \sum_{j=1}^{\infty} \lambda_j^2 (\hat{c}_j - c_j)^2 < \psi_j, \psi_j >_{\mathscr{H}_K}^2$$

$$= \sum_{j=1}^{\infty} (\hat{c}_j - c_j)^2.$$

We will prove the theorem using parts A.1-A.3 given below.

## A.1 ASYMPTOTIC ORDER FOR DETERMINISTIC ERROR $\bar{f}_{\infty,\lambda^\star} - f_0$.

Write $f_0(x) = \sum_{j=1}^{\infty} a_j \psi_j(x)$ and $f(x) = \sum_{j=1}^{\infty} c_j \psi_j(x)$, then we have

$$l_\infty(f) = \sigma^2 + \sum_{j=1}^{\infty}(c_j - a_j)^2.$$

By the orthogonality of the eigenfunctions, we have $\int_R \psi_j(x)^2 \pi(x)dx = 1$ and $\int_R \psi_j(x)\psi_l(x)\pi(x)dx = 0$, then we can derive

$$
\begin{aligned}
l_\infty(f) &= \sigma^2 + \mathbb{E}\{f(X) - f_0(X)\}^2 \\
&= \sigma^2 + \mathbb{E}\{\sum_{j=1}^{\infty} c_j \psi_j(X) - \sum_{l=1}^{\infty} a_l \psi_l(X)\}^2 \\
&= \sigma^2 + \mathbb{E}\{\sum_{j=1}^{\infty}\sum_{l=1}^{\infty}(c_j - a_j)(c_l - a_l)\psi_j(X)\psi_l(X)\} \\
&= \sigma^2 + \sum_{j=l}^{\infty}(c_j - a_j)^2\mathbb{E}\{\psi_j(X)^2\} + \sum_{j\neq l}^{\infty}(c_j - a_j)(c_l - a_l)\mathbb{E}\{\psi_j(X)\psi_l(X)\} \\
&= \sigma^2 + \sum_{j=l}^{\infty}(c_j - a_j)^2\int_R \psi_j(x)^2\pi(x)dx + \sum_{j\neq l}^{\infty}(c_j - a_j)(c_l - a_l)\int_R \psi_j(x)\psi_l(x)\pi(x)dx \\
&= \sigma^2 + \sum_{j=1}^{\infty}(c_j - a_j)^2.
\end{aligned}
$$

Then the corresponding objective function can be expressed as:

$$\bar{f}_{\infty,\lambda^\star}(c_j) = \arg\min\{Q_\infty(c_j)\} = \arg\min\{\sigma^2 + \sum_{j=l}^{\infty}(c_j - a_j)^2 + \lambda^\star \sum_{j=1}^{\infty}\frac{c_j^2}{\lambda_j}\}.$$

We then take derivative w.r.t. $c_j$ and obtain

$$Q'_\infty(c_j) = 2c_j - 2a_j + 2\lambda^\star\lambda_j^{-1}c_j.$$

It follows that the minimizer of the above objective function can be written as:

$$\bar{f}_{\infty,\lambda^\star}(x) = \sum_{j=1}^{\infty}\bar{c}_j\psi_j(x) = \sum_{j=1}^{\infty}\frac{a_j}{1 + \lambda^\star\lambda_j^{-1}}\psi_j(x),$$

where $\bar{c}_j = a_j/(1 + \lambda^\star\lambda_j^{-1})$.

To bound the deterministic error, assume $a_j^2\lambda_j^{-1} = j^{-a}$ with $a > 1$ and $\lambda_j \asymp j^{-k}$ ($k > 1$), so $a_j^2 \asymp j^{-(a+k)}$.:

$$\|\bar{f}_{\infty,\lambda^\star} - f_0\|^2 := \sum_{j=1}^{\infty}(\bar{c}_j - a_j)^2 = \sum_{j=1}^{\infty}\left(\frac{\lambda^\star\lambda_j^{-1}}{1 + \lambda^\star\lambda_j^{-1}}\right)^2 a_j^2 = \sum_{j=1}^{\infty}\left(\frac{\lambda^\star}{\lambda_j + \lambda^\star}\right)^2 a_j^2.$$

Let $J$ solve $\lambda_J \asymp \lambda^\star$ so $J \asymp \lambda^{\star -1/k}$. Split the sum at $J$:

$$\sum_{j\leq J}\left(\frac{\lambda^\star}{\lambda_j + \lambda^\star}\right)^2 a_j^2 \lesssim \lambda^{\star 2}\sum_{j\leq J}\frac{a_j^2}{\lambda_j^2} \asymp \lambda^{\star 2}\sum_{j\leq J}j^{k-a} \asymp \lambda^{\star 2}J^{1+k-a} \asymp \lambda^{\star\frac{a+k-1}{k}},$$

$$\sum_{j>J}\left(\frac{\lambda^\star}{\lambda_j + \lambda^\star}\right)^2 a_j^2 \lesssim \sum_{j>J}a_j^2 \asymp \sum_{j>J}j^{-(a+k)} \asymp J^{-(a+k-1)} \asymp \lambda^{\star\frac{a+k-1}{k}}.$$

Therefore,

$$\|\bar{f}_{\infty,\lambda^\star} - f_0\|^2 \asymp \lambda^{\star\frac{a+k-1}{k}} \qquad (a > 1,\ k > 1).$$

## A.2 Asymptotic Order for Stochastic Error $\hat{f}_{N,\lambda^\star} - \bar{f}_{\infty,\lambda^\star}$.

Recall that

$$l_N(f) = \frac{1}{N}\sum_{i=1}^{N}\{y_i - f(X_i)\}^2,$$

and the objective function for $\hat{f}_{N,\lambda^\star}$ can be written as:

$$\hat{f}_{N,\lambda^\star} = \arg\min_f\{Q_N(f)\} = \arg\min_f\{l_N(f) + \lambda^\star\|f\|_{\mathscr{H}_K}^2\}.$$

Note that the functional derivatives of the objective function with respect to $f$ is stochastic, which leads to a stochastic denominator in the solution of the above objective function, and it is not straightforward to handle a stochastic denominator. To avoid such difficulty, we define an intermediate quantity

$$\tilde{f}_{N,\lambda^\star} = \bar{f}_{\infty,\lambda^\star} - \frac{1}{2}G_{\lambda^\star}^{-1}Dl_{N,\lambda^\star}(\bar{f}_{\infty,\lambda^\star})$$

where $G_{\lambda^\star} = \frac{1}{2}D^2 l_{\infty,\lambda^\star}(\bar{f}_{\infty,\lambda^\star})$. Then we could write:

$$\hat{f}_{N,\lambda^\star} - \bar{f}_{\infty,\lambda^\star} = (\hat{f}_{N,\lambda^\star} - \tilde{f}_{N,\lambda^\star}) + (\tilde{f}_{N,\lambda^\star} - \bar{f}_{\infty,\lambda^\star}). \tag{3}$$

To find the orders of the above two terms, we need the functional derivatives given below. For functions $\eta, g \in \mathscr{H}_K$ and define the dot product $\eta \cdot g = <\eta, g>_{\mathscr{H}_K}$

$$
\begin{aligned}
Dl_N(f) \cdot \eta &= \frac{d}{dh}\frac{1}{N}\sum_{i=1}^{N}\{y_i - \langle f + h\eta, K(X_i,\cdot)\rangle_{\mathscr{H}_K}\}^2\big|_{h=0} \\
&= \frac{d}{dh}\frac{1}{N}\sum_{i=1}^{N}\{y_i - \langle f, K(X_i,\cdot)\rangle_{\mathscr{H}_K} - h\langle\eta, K(X_i,\cdot)\rangle_{\mathscr{H}_K}\}^2\big|_{h=0} \\
&= -\frac{2}{N}\sum_{i=1}^{N}\langle\eta, K(X_i,\cdot)\rangle_{\mathscr{H}_K}\{y_i - \langle f, K(X_i,\cdot)\rangle_{\mathscr{H}_K}\} \\
&= -\frac{2}{N}\sum_{i=1}^{N}\eta(X_i)\{y_i - f(X_i)\}. \\
Dl_\infty(f) \cdot \eta &= -2\int\eta(x)(f_0(x) - f(x))\pi(x)dx \\
&= -2\int\langle\eta, K(x,\cdot)\rangle_{\mathscr{H}_K}\langle f_0 - f, K(x,\cdot)\rangle_{\mathscr{H}_K}\pi(x)dx \\
D^2 l_N(f) \cdot \eta \cdot g &= \frac{2}{N}\sum_{i=1}^{N}\langle\eta, K(X_i,\cdot)\rangle_{\mathscr{H}_K}\langle g, K(X_i,\cdot)\rangle_{\mathscr{H}_K} \\
&= \frac{2}{N}\sum_{i=1}^{N}\langle\langle\eta, K(X_i,\cdot)\rangle_{\mathscr{H}_K}K(X_i,\cdot), g\rangle_{\mathscr{H}_K}. \\
D^2 l_\infty(f) \cdot \eta \cdot g &= 2\langle\int\langle\eta, K(x,\cdot)\rangle_{\mathscr{H}_K}K(x,\cdot)\pi(x)dx, g\rangle_{\mathscr{H}_K} \\
&= 2\sum_{j=1}^{\infty}\lambda_j^2\langle\eta, \psi_j\rangle_{\mathscr{H}_K}\langle g, \psi_j\rangle_{\mathscr{H}_K}. \\
D\|f\|_{\mathscr{H}_K}^2 \cdot \eta &= 2\sum_{j=1}^{\infty}\lambda_j\langle f, \psi_j\rangle_{\mathscr{H}_K}\langle\eta, \psi_j\rangle_{\mathscr{H}_K} \quad\text{and} \\
D^2\|f\|_{\mathscr{H}_K}^2 \cdot \eta \cdot g &= 2\sum_{j=1}^{\infty}\lambda_j\langle g, \psi_j\rangle_{\mathscr{H}_K}\langle\eta, \psi_j\rangle_{\mathscr{H}_K}.
\end{aligned}
$$

### A.2.1 EVALUATE THE ORDER OF $\tilde{f}_{N,\lambda^\star} - \bar{f}_{\infty,\lambda^\star}$.

Starting from the second term $\tilde{f}_{N,\lambda^\star} - \bar{f}_{\infty,\lambda^\star}$ in equation (3), because the functional derivatives to the penalty terms $\lambda^\star \|f\|^2_{\mathscr{H}_K}$ are the same for $l_{N,\lambda^\star}(\bar{f})$ and $l_{\infty,\lambda^\star}(\bar{f})$, and $\bar{f}$ is the minimizer of $l_{\infty,\lambda^\star}(f)$ that satisfies that $Dl_{\infty,\lambda^\star}(\bar{f}) = 0$. So, we have

$$Dl_{N,\lambda^\star}(\bar{f}) = Dl_{N,\lambda^\star}(\bar{f}) - Dl_{\infty,\lambda^\star}(\bar{f}) = Dl_N(\bar{f}) - Dl_\infty(\bar{f}).$$

For any function $\eta$, we have

$$\mathbb{E}\{Dl_{N,\lambda^\star}(\bar{f}) \cdot \eta\}^2 = \mathbb{E}\{Dl_N(\bar{f}) \cdot \eta - Dl_\infty(\bar{f}) \cdot \eta\}^2$$

$$= \mathbb{E}\left[ -\frac{2}{N} \sum_{i=1}^N \{y_i - \bar{f}(X_i)\}\eta(X_i) + 2\int \{\bar{f}(x) - f_0(x)\}\eta(x)\pi(x)dx \right]^2$$

$$= \frac{4}{N^2} \mathbb{E}\left[ \sum_{i=1}^N \left\{ (y_i - \bar{f}(X_i))\eta(X_i) - \mathbb{E}[(\bar{f}(X) - f_0(X))\eta(X)] \right\} \right]^2$$

$$= \frac{4}{N} \mathrm{Var}\left[ \{y - \bar{f}(X)\}\eta(X) \right] \leq \frac{4}{N} \mathbb{E}\left[ \{y - \bar{f}(X)\}\eta(X) \right]^2 \asymp \frac{4}{N}.$$

By the definition of $G_{\lambda^\star}$, we have

$$G_{\lambda^\star} \cdot \eta \cdot g = \frac{1}{2}D^2_{\infty,\lambda^\star} \cdot \eta \cdot g = \frac{1}{2}Dl^2_\infty(f) \cdot \eta \cdot g + \frac{1}{2}D^2\|f\|^2_{\mathscr{H}_{\mathscr{K}}} \cdot \eta \cdot g$$

$$= \sum_{j=1}^\infty \lambda_j^2 \langle \eta, \psi_j \rangle_{\mathscr{H}_K} \langle g, \psi_j \rangle_{\mathscr{H}_K} + \sum_{j=1}^\infty \lambda^\star \lambda_j < \eta, \psi_j >_{\mathscr{H}_K} < g, \psi_j >_{\mathscr{H}_K}$$

$$= \sum_{j=1}^\infty \lambda_j^2 (1 + \lambda_j^{-1}\lambda^\star)\langle g, \psi_j \rangle_{\mathscr{H}_K} \langle \eta, \psi_j \rangle_{\mathscr{H}_K}.$$

Let $\eta = \psi_m$ gives

$$\langle G_{\lambda^\star} g, \psi_m \rangle_{\mathscr{H}_K} = (\lambda_m + \lambda^\star) \langle g, \psi_m \rangle_{\mathscr{H}_K},$$

which leads to

$$\langle G_{\lambda^\star}^{-1} g, \psi_m \rangle_{\mathscr{H}_K} = (\lambda_m + \lambda^\star)^{-1} \langle g, \psi_m \rangle_{\mathscr{H}_K}.$$

Then, we can bound the second term in (3) by:

$$\mathbb{E}\|\tilde{f}_{N,\lambda^\star} - \bar{f}_{\infty,\lambda^\star}\|^2 = \mathbb{E}\left\| \tfrac{1}{2}G_{\lambda^\star}^{-1} Dl_{N,\lambda^\star}(\bar{f}_{\infty,\lambda^\star}) \right\|^2$$

$$= \tfrac{1}{4}\mathbb{E}\left\{ \sum_{j=1}^\infty \lambda_j^2 \left\langle G_{\lambda^\star}^{-1} Dl_{N,\lambda^\star}(\bar{f}_{\infty,\lambda^\star}), \psi_j \right\rangle^2_{\mathscr{H}_K} \right\}$$

$$= \tfrac{1}{4}\mathbb{E}\left\{ \sum_{j=1}^\infty \lambda_j^2 \frac{1}{(\lambda_j + \lambda^\star)^2} \left\langle Dl_{N,\lambda^\star}(\bar{f}_{\infty,\lambda^\star}), \psi_j \right\rangle^2_{\mathscr{H}_K} \right\}$$

$$= \tfrac{1}{4}\sum_{j=1}^\infty \frac{\lambda_j^2}{(\lambda_j + \lambda^\star)^2} \mathbb{E}\left\{ \left\langle Dl_{N,\lambda^\star}(\bar{f}_{\infty,\lambda^\star}), \psi_j \right\rangle^2_{\mathscr{H}_K} \right\}$$

$$\lesssim \frac{1}{N}\sum_{j=1}^\infty \frac{\lambda_j^2}{(\lambda_j + \lambda^\star)^2} \qquad \text{(using } \mathbb{E}\{\langle Dl_{N,\lambda^\star}(\bar{f}), \psi_j \rangle^2\} \asymp N^{-1})$$

$$\asymp \frac{1}{N}\int_1^\infty \frac{1}{(1 + \lambda^\star j^k)^2}\, dj \qquad \text{(since } \lambda_j \asymp j^{-k}, k > 1)$$

$$\asymp \frac{1}{N} \lambda^{\star -1/k}.$$

### A.2.2 EVALUATE THE ORDER OF $\hat{f}_{N,\lambda^\star} - \tilde{f}_{N,\lambda^\star}$.

Next we need to find the stochastic order of the second term $\hat{f}_{N,\lambda^\star} - \tilde{f}_{N,\lambda^\star}$ in the expression (3). For brevity, write

$$\hat{f} := \hat{f}_{N,\lambda^\star}, \qquad \tilde{f} := \tilde{f}_{N,\lambda^\star}, \qquad \bar{f} := \bar{f}_{\infty,\lambda^\star}.$$

Note that $\hat{f}$ is the solution of the following first order equation

$$Dl_{N,\lambda^\star}(\hat{f}) = Dl_{N,\lambda^\star}(\bar{f}) + D^2 l_{N,\lambda^\star}(\bar{f}) \cdot (\hat{f} - \bar{f}) = 0,$$

and by the definition of $\tilde{f}$, we have

$$Dl_{N,\lambda^\star}(\bar{f}) + D^2 l_{\infty,\lambda^\star}(\bar{f}) \cdot (\tilde{f} - \bar{f}) = 0.$$

From the above two equations, we can find:

$$D^2 l_{N,\lambda^\star}(\bar{f}) \cdot (\hat{f} - \bar{f}) = D^2 l_{\infty,\lambda^\star}(\bar{f}) \cdot (\hat{f} - \bar{f}).$$

Then we can write:

$$
\begin{aligned}
D^2 l_{\infty,\lambda^\star}(\bar{f}) \cdot (\hat{f} - \tilde{f}) &= D^2 l_{\infty,\lambda^\star}(\bar{f}) \cdot (\hat{f} - \bar{f}) + D^2 l_{\infty,\lambda^\star}(\bar{f}) \cdot (\bar{f} - \tilde{f}) \\
&= D^2 l_{\infty,\lambda^\star}(\bar{f}) \cdot (\hat{f} - \bar{f}) - D^2 l_{N,\lambda^\star}(\bar{f}) \cdot (\hat{f} - \bar{f}) \\
&= D^2 l_\infty(\bar{f}) \cdot (\hat{f} - \bar{f}) - D^2 l_N(\bar{f}) \cdot (\hat{f} - \bar{f}).
\end{aligned}
$$

Using the definition of $G_{\lambda^\star}$, we have

$$\hat{f} - \tilde{f} = \frac{1}{2} G_{\lambda^\star}^{-1} D^2 l_{\infty,\lambda^\star}(\bar{f}) \cdot (\hat{f} - \tilde{f}) = \frac{1}{2} G_{\lambda^\star}^{-1} \Big\{ D^2 l_\infty(\bar{f}) \cdot (\hat{f} - \bar{f}) - D^2 l_N(\bar{f}) \cdot (\hat{f} - \bar{f}) \Big\}.$$

Using the similar steps in evaluating $\mathbb{E}\|\tilde{f} - \bar{f}\|^2$, we have

$$\|\hat{f} - \tilde{f}\|^2$$

$$= \left\| \tfrac{1}{2} G_{\lambda^\star}^{-1} \Big\{ D^2 l_\infty(\bar{f})(\hat{f} - \bar{f}) - D^2 l_N(\bar{f})(\hat{f} - \bar{f}) \Big\} \right\|^2$$

$$= \frac{1}{4} \sum_{j=1}^\infty \lambda_j^2 \left\langle G_{\lambda^\star}^{-1} \Big\{ D^2 l_\infty(\bar{f})(\hat{f} - \bar{f}) - D^2 l_N(\bar{f})(\hat{f} - \bar{f}) \Big\}, \psi_j \right\rangle_{\mathscr{H}_K}^2$$

$$= \frac{1}{4} \sum_{j=1}^\infty \lambda_j^2 \frac{1}{(\lambda_j + \lambda^\star)^2} \left\{ \langle D^2 l_\infty(\bar{f})(\hat{f} - \bar{f}), \psi_j \rangle_{\mathscr{H}_K} - \langle D^2 l_N(\bar{f})(\hat{f} - \bar{f}), \psi_j \rangle_{\mathscr{H}_K} \right\}^2$$

$$= \frac{1}{4} \sum_{j=1}^\infty \lambda_j^2 (1 + \lambda^\star \lambda_j^{-1})^{-2} \left\{ < D^2 l_\infty(\bar{f})(\hat{f} - \bar{f}), \psi_j >_{\mathscr{H}_{\mathscr{K}}} - < D^2 l_N(\bar{f})(\hat{f} - \bar{f}), \psi_j >_{\mathscr{H}_{\mathscr{K}}} \right\}^2$$

$$= \frac{1}{4} \sum_{j=1}^\infty \lambda_j^2 (1 + \lambda^\star \lambda_j^{-1})^{-2} \Big\{ 2 \sum_{l=1}^\infty \lambda_l^2 < \hat{f} - \bar{f}, \psi_l >_{\mathscr{H}_{\mathscr{K}}} < \psi_j, \psi_l >_{\mathscr{H}_{\mathscr{K}}}$$

$$\qquad\qquad - \frac{2}{N} \sum_{i=1}^N \sum_{l=1}^\infty \sum_{k=1}^\infty \lambda_l \lambda_k < \hat{f} - \bar{f}, \psi_l >_{\mathscr{H}_{\mathscr{K}}} < \psi_j, \psi_k >_{\mathscr{H}_{\mathscr{K}}} \psi_l(X_i) \psi_k(X_i) \Big\}^2$$

$$= \frac{1}{4} \sum_{j=1}^\infty \lambda_j^2 (1 + \lambda^\star \lambda_j^{-1})^{-2} \Big[ \frac{2}{N} \sum_{i=1}^N \sum_{l=1}^\infty \lambda_l < \hat{f} - \bar{f}, \psi_l >_{\mathscr{H}_{\mathscr{K}}} \mathbb{E}\{\psi_j(X_i)\psi_l(X_i)\}$$

$$\qquad\qquad - \frac{2}{N} \sum_{i=1}^N \sum_{l=1}^\infty \lambda_l < \hat{f} - \bar{f}, \psi_l >_{\mathscr{H}_{\mathscr{K}}} \psi_l(X_i)\psi_k(X_i) \Big]^2$$

$$= \frac{1}{4} \sum_{j=1}^\infty \lambda_j^2 (1 + \lambda^\star \lambda_j^{-1})^{-2} \Big[ \frac{2}{N} \sum_{l=1}^\infty \lambda_l^{\frac{1}{2}} < \hat{f} - \bar{f}, \psi_l >_{\mathscr{H}_{\mathscr{K}}} \lambda_l^{\frac{1}{2}} \Big\{ \sum_{i=1}^N \mathbb{E}\{\psi_j(X_i)\psi_l(X_i)\} - \sum_{i=1}^N \psi_j(X_i)\psi_l(X_i) \Big\} \Big]^2$$

$$\leq N^{-2} \sum_{j=1}^{\infty} \lambda_j^2 (1 + \lambda^\star \lambda_j^{-1})^{-2} \Big( \sum_{l=1}^{\infty} \lambda_l \langle \hat{f} - \bar{f}, \psi_l \rangle^2_{\mathscr{H}_K} \Big) \Big[ \sum_{l=1}^{\infty} \lambda_l \Big\{ \sum_{i=1}^{N} \mathbb{E}(\psi_j(X_i)\psi_l(X_i)) - \sum_{i=1}^{N} \psi_j(X_i)\psi_l(X_i) \Big\}^2 \Big]$$

$$= N^{-2} \sum_{j=1}^{\infty} \lambda_j^2 (1 + \lambda^\star \lambda_j^{-1})^{-2} \|\hat{f} - \bar{f}\|^2_{\mathscr{H}_K} \underbrace{\Big[ \sum_{l=1}^{\infty} \lambda_l \Big\{ \sum_{i=1}^{N} \mathbb{E}(\psi_j(X_i)\psi_l(X_i)) - \sum_{i=1}^{N} \psi_j(X_i)\psi_l(X_i) \Big\}^2 \Big]}_{\text{Term A}}.$$

For Term A, we can expand it as:

$$\sum_{l=1}^{\infty} \lambda_l \left[ \sum_{i=1}^{N} \Big\{ \mathbb{E}\Big(\psi_j(X_i)\psi_l(X_i)\Big) - \psi_j(X_i)\psi_l(X_i) \Big\} \right]^2$$

$$= \sum_{l=1}^{\infty} \lambda_l \left[ \sum_{i=1}^{N} \Big\{ \mathbb{E}\Big(\psi_j(X_i)\psi_l(X_i)\Big) - \psi_j(X_i)\psi_l(X_i) \Big\}^2 \right]$$

$$+ \sum_{l=1}^{\infty} \lambda_l \left[ \sum_{i \neq k}^{N} \Big\{ \mathbb{E}\Big(\psi_j(X_i)\psi_l(X_i)\Big) - \psi_j(X_i)\psi_l(X_i) \Big\} \Big\{ \mathbb{E}\Big(\psi_j(X_k)\psi_l(X_k)\Big) - \psi_j(X_k)\psi_l(X_k) \Big\} \right]$$

$$= \underbrace{\lambda_j \sum_{i=1}^{N} \Big\{ \mathbb{E}(\psi_j^2(X_i)) - \psi_j^2(X_i) \Big\}^2}_{\text{Term 1}}$$

$$+ \underbrace{\sum_{l=1}^{\infty} \lambda_l \sum_{i \neq k}^{N} \Big[ \mathbb{E}\{\psi_j(X_i)\psi_l(X_i)\} - \psi_j(X_i)\psi_l(X_i) \Big] \Big[ \mathbb{E}\{\psi_j(X_k)\psi_l(X_k)\} - \psi_j(X_k)\psi_l(X_k) \Big]}_{\text{Term 2}}.$$

To find the orders of Term 1 and Term 2, we evaluate the expectation of Term 1 and the variance of Term 2, which are given below:

$$\mathbb{E}\left[ \lambda_j \sum_{i=1}^{N} \Big\{ \mathbb{E}(\psi_j^2(X_i)) - \psi_j^2(X_i) \Big\}^2 \right]$$

$$= \mathbb{E}\left[ \lambda_j \sum_{i=1}^{N} \big\{ \psi_j^2(X_i) - 1 \big\}^2 + \sum_{l \neq j}^{\infty} \lambda_l \sum_{i=1}^{N} \big\{ \psi_j(X_i)\psi_l(X_i) \big\}^2 \right]$$

$$= \lambda_j N \, \mathbb{E}\big\{ \psi_j^2(X_i) - 1 \big\}^2 + N \mathbb{E}\Big\{ \sum_{l \neq j}^{\infty} \lambda_l \psi_j^2(X_i)\psi_l^2(X_i) \Big\}$$

$$\leq \lambda_j N \, \mathrm{Var}\{\psi_j^2(X)\} + N \Big[ \mathbb{E}\Big\{ \sum_{l=1}^{\infty} \lambda_l \psi_l^2(X) \Big\}^2 + \mathrm{Var}\Big\{ \sum_{l=1}^{\infty} \lambda_l \psi_l^2(X) \Big\} \Big]^{1/2} \big[ \mathbb{E}\{\psi_j^4(X)\} \big]^{1/2}$$

$$= \lambda_j N \, \mathrm{Var}\{\psi_j^2(X)\} + N \Big[ \mathbb{E}\{K(X,X)\}^2 + \mathrm{Var}\{K(X,X)\} \Big]^{1/2} \big[ \mathbb{E}\{\psi_j^4(X)\} \big]^{1/2}$$

$$\asymp N.$$

$$\mathrm{Var}\left[ \sum_{l=1}^{\infty} \lambda_l \sum_{i \neq k}^{N} \Big\{ \mathbb{E}[\psi_j(X_i)\psi_l(X_i)] - \psi_j(X_i)\psi_l(X_i) \Big\} \Big\{ \mathbb{E}[\psi_j(X_k)\psi_l(X_k)] - \psi_j(X_k)\psi_l(X_k) \Big\} \right]$$

$$\leq \sum_{l=1}^{\infty} \lambda_l^2 (N-1)^2 \, \mathbb{E}\Big[ \big\{ \mathbb{E}[\psi_j(X_i)\psi_l(X_i)] - \psi_j(X_i)\psi_l(X_i) \big\}^2 \Big] \mathbb{E}\Big[ \big\{ \mathbb{E}[\psi_j(X_k)\psi_l(X_k)] - \psi_j(X_k)\psi_l(X_k) \big\}^2 \Big]$$

$$+ \sum_{l \neq v}^{\infty} \lambda_l \lambda_v (N-1)^2 \, \mathrm{Var}\{\psi_j(X)\psi_l(X)\} \, \mathrm{Var}\{\psi_j(X)\psi_v(X)\}$$

$$= (N-1)^2 \sum_{l=1}^{\infty} \lambda_l^2 \operatorname{Var}^2\{\psi_j(X)\psi_l(X)\} + (N-1)^2 \sum_{l \neq v}^{\infty} \lambda_l \lambda_v \operatorname{Var}\{\psi_j(X)\psi_l(X)\} \operatorname{Var}\{\psi_j(X)\psi_v(X)\}$$

$$\leq (N-1)^2 \left[ \sum_{l=1}^{\infty} \lambda_l^2 \operatorname{Var}^2\{\psi_j(X)\psi_l(X)\} + \left[ \sum_{l=1}^{\infty} \lambda_l^2 \operatorname{Var}^2\{\psi_j(X)\psi_l(X)\} \right]^{1/2} \left[ \sum_{v=1}^{\infty} \lambda_v^2 \operatorname{Var}^2\{\psi_j(X)\psi_v(X)\} \right]^{1/2} \right]$$

$$\asymp 2(N-1)^2 \int_1^{\infty} l^{-2k} \, dl \asymp N^2.$$

Now we found that Term 1 is at the order of $N$ and Term 2 is at the order of $\sqrt{N^2} = N$, therefore we can write:

$$\|\hat{f}_{N,\lambda^\star} - \tilde{f}_{N,\lambda^\star}\|_{\mathscr{H}_{\mathscr{K}}}^2 = N^{-1} \sum_{j=1}^{\infty} \lambda_j^2 (1 + \lambda^\star \lambda_j^{-1})^{-2} \|\hat{f}_{N,\lambda^\star} - \bar{f}_{\infty,\lambda^\star}\|_{\mathscr{H}_{\mathscr{K}}}^2$$

$$= N^{-1} \int_1^{\infty} \lambda_j^2 (1 + \lambda^\star \lambda_j^{-1})^{-2} \|\hat{f}_{N,\lambda^\star} - \bar{f}_{\infty,\lambda^\star}\|_{\mathscr{H}_{\mathscr{K}}}^2$$

$$\asymp N^{-1} (\lambda^\star)^{-1/k} \|\hat{f}_{N,\lambda^\star} - \bar{f}_{\infty,\lambda^\star}\|_{\mathscr{H}_{\mathscr{K}}}^2.$$

Now, it follows that

$$\|\hat{f}_{N,\lambda^\star} - \tilde{f}_{N,\lambda^\star}\|_{\mathscr{H}_{\mathscr{K}}}^2 = O_p\Big( N^{-1}(\lambda^\star)^{-1/k} \|\hat{f}_{N,\lambda^\star} - \bar{f}_{\infty,\lambda^\star}\|_{\mathscr{H}_{\mathscr{K}}}^2 \Big).$$

If $N^{-1}(\lambda^\star)^{-1/k} \to 0$,

$$\|\hat{f}_{N,\lambda^\star} - \tilde{f}_{N,\lambda^\star}\|_{\mathscr{H}_{\mathscr{K}}}^2 = o_p\Big( \|\hat{f}_{N,\lambda^\star} - \bar{f}_{\infty,\lambda^\star}\|_{\mathscr{H}_{\mathscr{K}}} \Big) = o_p(1) \|\hat{f}_{N,\lambda^\star} - \bar{f}_{\infty,\lambda^\star}\|_{\mathscr{H}_{\mathscr{K}}}.$$

Observed that

$$\|\tilde{f}_{N,\lambda^\star} - \bar{f}_{\infty,\lambda^\star}\|_{\mathscr{H}_{\mathscr{K}}}^2 \geq \|\hat{f}_{N,\lambda^\star} - \bar{f}_{\infty,\lambda^\star}\|_{\mathscr{H}_{\mathscr{K}}}^2 - \|\hat{f}_{N,\lambda^\star} - \tilde{f}_{N,\lambda^\star}\|_{\mathscr{H}_{\mathscr{K}}}^2 = (1 - o_p(1)) \|\hat{f}_{N,\lambda^\star} - \bar{f}_{\infty,\lambda^\star}\|_{\mathscr{H}_{\mathscr{K}}}^2,$$

then

$$\|\hat{f}_{N,\lambda^\star} - \bar{f}_{\infty,\lambda^\star}\|_{\mathscr{H}_{\mathscr{K}}}^2 = O_p\Big( \|\tilde{f}_{N,\lambda^\star} - \bar{f}_{\infty,\lambda^\star}\|_{\mathscr{H}_{\mathscr{K}}} \Big) = O_p\big( N^{-1}(\lambda^\star)^{-1/k} \big).$$

### A.3 ASYMPTOTIC ORDER FOR MSE AND TUNING.

Combining the results in previous steps, we can express the order of MSE by:

$$\|\hat{f}_{N,\lambda^\star} - f_0\|^2 = O_p\big( N^{-1}(\lambda^\star)^{-1/k} + (\lambda^\star)^{\frac{a+k-1}{k}} \big).$$

To find the optimal order of $\lambda^\star$, set $M(d) = N^{-1}d^{-1/k} + d^{\frac{a+k-1}{k}}$. Then

$$M'(d) = -\frac{1}{k}N^{-1}d^{-1/k-1} + \frac{a+k-1}{k} d^{\frac{a+k-1}{k}-1} = 0,$$

which yields

$$d^{\frac{a+k}{k}} \asymp N^{-1} \implies \lambda^\star \asymp N^{-\frac{k}{a+k}}.$$

Also check

$$N^{-1}(\lambda^\star)^{-1/k} = N^{-1}\big( N^{-k/(a+k)} \big)^{-1/k} = N^{-\frac{a+k-1}{a+k}} \to 0 \quad (a > 1, \ k > 1).$$

Therefore,

$$\|\hat{f}_{N,\lambda^\star} - f_0\|^2 = O_p\big( N^{-1+\frac{1}{a+k}} \big).$$

This completes the proof of Theorem 1.

# B  PROOF OF THEOREM 2

For any given $x_0$, the proposed resampling weights and any sample $X_i \in \mathscr{C}_l$ where $\mathscr{C}_l$ is the l-th cluster with its corresponding center $C_l$ and its cluster size $N_l$, the weight/probability of $X_i$ to be selected is:

$$\omega_i(x_0) = N_l^{-1} \omega_{x_0,l,C}.$$

So we have:

$$\sum_{i=1}^{N} \omega_i(x_0) = \sum_{l=1}^{L} \sum_{i \in \mathscr{C}_l} \frac{1}{N_l} \omega_{x_0,l,C} = \sum_{l=1}^{L} \omega_{x_0,l,C} = 1.$$

Denote the indicies of the resampling subset of size $n$ for the given $x_0$ is $\mathscr{S}(x_0) = \{i_1^*, \ldots, i_n^*\} \subset \{1, \ldots, N\}$, with replacement using probabilities $\{\omega_i(x_0)\}$. Let $I_i(x_0)$ be the count of how many times index $i$ appears in $\mathscr{S}(x_0)$ so $\mathbb{E}[I_i(x_0)] = n\,\omega_i(x_0)$. Define

$$l_{n,x_0}(f) = \frac{1}{n} \sum_{i \in \mathscr{S}(x_0)} (y_i - f(X_i))^2 = \frac{1}{n} \sum_{i=1}^{N} I_i(x_0)\left(y_i - f(X_i)\right)^2.$$

Recall that the proposed estimator of $f(x_0)$ is $\hat{f}_{n,\tilde{\lambda}^\star,x_0}(x_0)$, which is given by:

$$\hat{f}_{n,\tilde{\lambda}^\star,x_0} = \arg \min_{f \in \mathscr{H}_K} \left\{ \frac{1}{n} \sum_{i \in \mathscr{S}(x_0)} (y_i - f(X_i))^2 + \tilde{\lambda}^\star \|f\|_{\mathscr{H}_K}^2 \right\}.$$

Define the functional $\hat{f}_{n,\tilde{\lambda}^\star}$ of the proposed estimator by collecting the estimators at all $x_0 \in \mathcal{X}$ together $\hat{f}_{n,\tilde{\lambda}^\star} = \left\{ \hat{f}_{n,\tilde{\lambda}^\star,x_0}(x_0) : x_0 \in \mathcal{X} \right\}$, where $\mathcal{X}$ is the space of the predictor/feature $x$. Define the norm:

$$\|\hat{f}_{n,\tilde{\lambda}^\star} - f\|^2 = \int <\hat{f}_{n,\tilde{\lambda}^\star,x_0} - f, K(x_0, \cdot) >_{\mathscr{H}_K}^2 \pi(x_0)dx_0$$

$$= \int <\hat{f}_{n,\tilde{\lambda}^\star,x_0} - f, \sum_{j=1}^{\infty} \lambda_j \psi_j(\cdot)\psi_j(x_0) >_{\mathscr{H}_K}^2 \pi(x_0)dx_0$$

$$= \int (\sum_{j=1}^{\infty} \lambda_j <\hat{f}_{n,\tilde{\lambda}^\star,x_0} - f, \psi_j(\cdot) >_{\mathscr{H}_K} \psi_j(x_0))^2 \pi(x_0)dx_0.$$

Here $l_\infty(f)$ is the limit of $l_n(f)$, with conditioning made explicit:

$$l_\infty(f) := \mathbb{E}\Big[ \frac{1}{n} \sum_{i \in \mathscr{S}(x_0)} \{y_i - f(X_i)\}^2 \Big]$$

$$= \mathbb{E}\Big[ \mathbb{E}\Big\{ \frac{1}{n} \sum_{i=1}^{N} I_i(x_0)(y_i - f(X_i))^2 \,\Big|\, \{X_i\}_{i=1}^{N} \Big\} \Big]$$

$$= \mathbb{E}\Big[ \frac{1}{n} \sum_{i=1}^{N} \mathbb{E}\big(I_i(x_0) \mid \{X_i\}_{i=1}^{N}\big)(y_i - f(X_i))^2 \Big]$$

$$= \mathbb{E}\Big[ \sum_{i=1}^{N} \omega_i(x_0)(y_i - f(X_i))^2 \Big] \qquad \big(\mathbb{E}[I_i(x_0) \mid \{X_i\}] = n\,\omega_i(x_0)\big)$$

$$= \mathbb{E}\Big[ \sum_{i=1}^{N} \omega_i(x_0)\{\sigma^2 + (f(X_i) - f_0(X_i))^2\} \Big]$$

$$= \sigma^2 + \mathbb{E}\Big[ \sum_{i=1}^{N} \omega_i(x_0)\big\{ \langle f - f_0, K(X_i, \cdot)\rangle_{\mathscr{H}_K} \big\}^2 \Big].$$

To evaluate the MSE of our proposed method, we evaluate the deterministic error and stochastic error separately by decomposing it in the following way:

$$\hat{f}_{n,\tilde{\lambda}^\star,x_0} - f_0 = (\hat{f}_{n,\tilde{\lambda}^\star,x_0} - \bar{f}_{\infty,\tilde{\lambda}^\star,x_0}) + (\bar{f}_{\infty,\tilde{\lambda}^\star,x_0} - f_0),$$

where $\bar{f}_{\infty,\tilde{\lambda}^\star,x_0}$ is the solution of the following objective function

$$\bar{f}_{\infty,\tilde{\lambda}^\star,x_0} = \arg\min \left\{ l_\infty(f) + \tilde{\lambda}^\star \|f\|_{\mathscr{H}_K}^2 \right\}.$$

First, we evaluate the functional derivatives for the empirical loss $l_{n,x_0}(f) = \frac{1}{n}\sum_{i\in\mathscr{S}(x_0)}\{y_i - f(X_i)\}^2$ and for its population limit $l_{\infty,x_0}(f) = \sigma^2 + \mathbb{E}\Big[\sum_{i=1}^N \omega_i(x_0)\{f(X_i) - f_0(X_i)\}^2\Big]$.

For any $\eta, g \in \mathscr{H}_K$, the derivatives of $l_{\infty,x_0}$ are

$$Dl_{\infty,x_0}(f)\cdot\eta = \frac{d}{dh}\, l_{\infty,x_0}(f+h\eta)\Big|_{h=0}$$

$$= 2\,\mathbb{E}\Big[\sum_{i=1}^N \omega_i(x_0)\,\eta(X_i)\,\{f(X_i) - f_0(X_i)\}\Big]$$

$$= 2\,\mathbb{E}\Big\{\sum_{i=1}^N \omega_i(x_0)\,\langle\eta, K(X_i,\cdot)\rangle_{\mathscr{H}_K}\,\langle f - f_0, K(X_i,\cdot)\rangle_{\mathscr{H}_K}\Big\}.$$

and

$$D^2 l_{\infty,x_0}(f)\cdot\eta\cdot g = 2\,\mathbb{E}\Big\{\sum_{i=1}^N \omega_i(x_0)\,\langle\eta, K(X_i,\cdot)\rangle_{\mathscr{H}_K}\,\langle g, K(X_i,\cdot)\rangle_{\mathscr{H}_K}\Big\}$$

$$= 2\sum_{j=1}^\infty \lambda_j^2\,\langle\eta, \psi_j\rangle_{\mathscr{H}_K}\,\langle g, \psi_j\rangle_{\mathscr{H}_K}\,\underbrace{\mathbb{E}\Big\{\sum_{i=1}^N \omega_i(x_0)\,\psi_j(X_i)^2\Big\}}_{=:\,\omega_{x_0,j}}.$$

For the empirical loss $l_{n,x_0}$,

$$Dl_{n,x_0}(f)\cdot\eta = \frac{d}{dh}\frac{1}{n}\sum_{i\in\mathscr{S}(x_0)}\big(y_i - \langle f+h\eta, K(X_i,\cdot)\rangle_{\mathscr{H}_K}\big)^2\Big|_{h=0}$$

$$= -\frac{2}{n}\sum_{i\in\mathscr{S}(x_0)}\langle\eta, K(X_i,\cdot)\rangle_{\mathscr{H}_K}\big(y_i - \langle f, K(X_i,\cdot)\rangle_{\mathscr{H}_K}\big),$$

and

$$D^2 l_{n,x_0}(f)\cdot\eta\cdot g = \frac{2}{n}\sum_{i\in\mathscr{S}(x_0)}\langle\eta, K(X_i,\cdot)\rangle_{\mathscr{H}_K}\,\langle g, K(X_i,\cdot)\rangle_{\mathscr{H}_K}$$

$$= \frac{2}{n}\sum_{i\in\mathscr{S}(x_0)}\sum_{j,k\geq 1}\lambda_j\lambda_k\,\langle\eta, \psi_j\rangle_{\mathscr{H}_K}\,\langle g, \psi_k\rangle_{\mathscr{H}_K}\,\psi_j(X_i)\psi_k(X_i).$$

For the RKHS penalty,

$$D\|f\|_{\mathscr{H}_K}^2\cdot\eta = 2\,\langle f, \eta\rangle_{\mathscr{H}_K},$$

$$D^2\|f\|_{\mathscr{H}_K}^2\cdot\eta\cdot g = 2\,\langle\eta, g\rangle_{\mathscr{H}_K} = 2\sum_{j=1}^\infty \lambda_j\,\langle\eta, \psi_j\rangle_{\mathscr{H}_K}\,\langle g, \psi_j\rangle_{\mathscr{H}_K}.$$

To evaluate the stochastic error, we use a similar method in the proof of Theorem 1 by defining an intermediate quantity

$$\tilde{f}_{n,\tilde{\lambda}^\star,x_0} = \bar{f}_{\infty,\tilde{\lambda}^\star,x_0} - \frac{1}{2}\,G_{\tilde{\lambda}^\star,x_0}^{-1}\,Dl_{n,x_0}(\bar{f}_{\infty,\tilde{\lambda}^\star,x_0}),$$

where the local operator at $x_0$ is

$$G_{\tilde{\lambda}^\star, x_0} = \frac{1}{2} D^2 l_{\infty, x_0}(\bar{f}_{\infty, \tilde{\lambda}^\star, x_0}) + \tilde{\lambda}^\star D^2 \|f\|_{\mathscr{H}_K}^2.$$

Then, we define the functional

$$\tilde{f}_{n, \tilde{\lambda}^\star} = \{\tilde{f}_{n, \tilde{\lambda}^\star, x_0} : x_0 \in \mathcal{X}\}.$$

Finally, we decompose the estimation error as

$$\hat{f}_{n, \tilde{\lambda}^\star} - \bar{f}_{\infty, \tilde{\lambda}^\star} = (\hat{f}_{n, \tilde{\lambda}^\star} - \tilde{f}_{n, \tilde{\lambda}^\star}) + (\tilde{f}_{n, \tilde{\lambda}^\star} - \bar{f}_{\infty, \tilde{\lambda}^\star}).$$

### B.1 ASYMPTOTIC ORDER FOR STOCHASTIC ERROR.

To figure out the order for stochatic error, we finish the proof in 2 steps.

#### B.1.1 STEP 1: EVALUATE THE ORDER OF $\tilde{f}_{n, \tilde{\lambda}^\star} - \bar{f}_{\infty, \tilde{\lambda}^\star}$.

Assume that $\omega_{x_0, j} = \mathbb{E}\{\sum_{i=1}^N \omega_i(x_0)\psi_j^2(X_i)\}$. To this end, we first obtain the eigenvalues of the operator $G_{\tilde{\lambda}^\star}$.

$$G_{\tilde{\lambda}^\star, x_0} \cdot \eta \cdot g = \frac{1}{2} D^2 l_{\infty, x_0}(\bar{f}_{\infty, \tilde{\lambda}^\star, x_0}) \cdot \eta \cdot g + \tilde{\lambda}^\star D^2 \|f\|_{\mathscr{H}_{\mathscr{K}}}^2 \cdot \eta \cdot g$$

$$= \sum_{j=1}^\infty \lambda_j^2 < \eta, \psi_j >_{\mathscr{H}_K} < g, \psi_j >_{\mathscr{H}_K} \mathbb{E}\{\sum_{i=1}^N \omega_i(x_0)\psi_j^2(X_i)\}$$

$$+ \sum_{j=1}^\infty \tilde{\lambda}^\star \lambda_j < \eta, \psi_j >_{\mathscr{H}_K} < g, \psi_j >_{\mathscr{H}_K}$$

$$= \sum_{j=1}^\infty \omega_{x_0, j} \lambda_j^2 < \eta, \psi_j >_{\mathscr{H}_K} < g, \psi_j >_{\mathscr{H}_K} + \sum_{j=1}^\infty \tilde{\lambda}^\star \lambda_j < \eta, \psi_j >_{\mathscr{H}_K} < g, \psi_j >_{\mathscr{H}_K}$$

$$= \sum_{j=1}^\infty (\omega_{x_0, j} \lambda_j^2 + \tilde{\lambda}^\star \lambda_j) < \eta, \psi_j >_{\mathscr{H}_K} < g, \psi_j >_{\mathscr{H}_K}$$

$$= \sum_{j=1}^\infty \lambda_j^2 (\omega_{x_0, j} + \tilde{\lambda}^\star \lambda_j^{-1}) < \eta, \psi_j >_{\mathscr{H}_K} < g, \psi_j >_{\mathscr{H}_K}.$$

Let $\eta = \psi_m$ gives

$$\langle G_{\tilde{\lambda}^\star, x_0} g, \psi_m \rangle_{\mathscr{H}_K} = \lambda_m (\omega_{x_0, m} + \tilde{\lambda}^\star \lambda_m^{-1}) \langle g, \psi_m \rangle_{\mathscr{H}_K}$$

which leads to

$$\langle G_{\tilde{\lambda}^\star, x_0}^{-1} g, \psi_m \rangle_{\mathscr{H}_K} = \lambda_m^{-1} (\omega_{x_0, m} + \tilde{\lambda}^\star \lambda_m^{-1})^{-1} \langle g, \psi_m \rangle_{\mathscr{H}_K}.$$

Using the expression of the norm and the expression of the operator $G_{\tilde{\lambda}^\star, x_0}$ and assuming $\lambda_j \asymp j^{-k}$ with $k > 1$, and $\omega_{x_0, j} \asymp j^\beta$ uniformly in $x_0$ for some $\beta < k$ (so that $k - \beta > 0$), we have the following:

$$\|\tilde{f}_{n, \tilde{\lambda}^\star} - \bar{f}_{\infty, \tilde{\lambda}^\star}\|^2 = \frac{1}{4} \int \left(\sum_{j=1}^\infty \lambda_j < G_{\tilde{\lambda}^\star, x_0}^{-1} Dl_{n, \tilde{\lambda}^\star}(\bar{f}_{\infty, \tilde{\lambda}^\star, x_0}), \psi_j >_{\mathscr{H}_K} \psi_j(x_0)\right)^2 \pi(x_0) \, dx_0$$

$$= \frac{1}{4} \int \left[\left\{\sum_{j=1}^\infty \lambda_j \lambda_j^{-1} (\omega_{x_0, j} + \tilde{\lambda}^\star \lambda_j^{-1})^{-1} < Dl_{n, \tilde{\lambda}^\star}(\bar{f}_{\infty, \tilde{\lambda}^\star, x_0}), \psi_j >_{\mathscr{H}_K} \psi_j(x_0)\right\}^2\right] \pi(x_0) \, dx_0$$

$$= \frac{1}{4} \int \left[ \left\{ \sum_{j=1}^{\infty} (\omega_{x_0,j} + \tilde{\lambda}^\star \lambda_j^{-1})^{-1} < Dl_{n,\tilde{\lambda}^\star}(\bar{f}_{\infty,\tilde{\lambda}^\star,x_0}), \psi_j >_{\mathscr{H}_K} \psi_j(x_0) \right\}^2 \right] \pi(x_0) \, dx_0$$

$$= \frac{1}{4} \int \sum_{j=1}^{\infty} (\omega_{x_0,j} + \tilde{\lambda}^\star \lambda_j^{-1})^{-2} < Dl_{n,\tilde{\lambda}^\star}(\bar{f}_{\infty,\tilde{\lambda}^\star,x_0}), \psi_j >_{\mathscr{H}_K}^2 \psi_j^2(x_0) \pi(x_0) \, dx_0$$

$$+ \frac{1}{4} \int \sum_{j \neq l}^{\infty} (\omega_{x_0,j} + \tilde{\lambda}^\star \lambda_j^{-1})^{-1} (\omega_{x_0,l} + \tilde{\lambda}^\star \lambda_l^{-1})^{-1} < Dl_{n,\tilde{\lambda}^\star}(\bar{f}_{\infty,\tilde{\lambda}^\star,x_0}), \psi_j >_{\mathscr{H}_K}$$

$$\cdot < Dl_{n,\tilde{\lambda}^\star}(\bar{f}_{\infty,\tilde{\lambda}^\star,x_0}), \psi_l >_{\mathscr{H}_K} \psi_j(x_0)\psi_l(x_0)\pi(x_0) \, dx_0$$

$$= \frac{1}{4} \int \sum_{j=1}^{\infty} \frac{1}{(j^\beta + \tilde{\lambda}^\star j^k)^2} < Dl_{n,\tilde{\lambda}^\star}(\bar{f}_{\infty,\tilde{\lambda}^\star,x_0}), \psi_j >_{\mathscr{H}_K}^2 \psi_j^2(x_0) \, \pi(x_0) \, dx_0$$

$$+ \frac{1}{4} \int \sum_{j \neq l}^{\infty} \frac{1}{(j^\beta + \tilde{\lambda}^\star j^k)} \frac{1}{(l^\beta + \tilde{\lambda}^\star l^k)} < Dl_{n,\tilde{\lambda}^\star}(\bar{f}_{\infty,\tilde{\lambda}^\star,x_0}), \psi_j >_{\mathscr{H}_K} < Dl_{n,\tilde{\lambda}^\star}(\bar{f}_{\infty,\tilde{\lambda}^\star,x_0}), \psi_l >_{\mathscr{H}_K}$$

$$\times \psi_j(x_0)\psi_l(x_0) \, \pi(x_0) \, dx_0$$

$$\asymp \frac{1}{4} \sum_{j=1}^{\infty} \frac{1}{\left( j^\beta + \tilde{\lambda}^\star j^k \right)^2} \mathbb{E}\{ \langle Dl_{n,\tilde{\lambda}^\star}(\bar{f}_{\infty,\tilde{\lambda}^\star,x_0}), \psi_j \rangle_{\mathscr{H}_K}^2 \} \int \psi_j^2(x_0) \, \pi(x_0) \, dx_0$$

$$+ \frac{1}{4} \sum_{j \neq l}^{\infty} \frac{1}{\left( j^\beta + \tilde{\lambda}^\star j^k \right)} \frac{1}{\left( l^\beta + \tilde{\lambda}^\star l^k \right)} \mathbb{E}\{ \langle Dl_{n,\tilde{\lambda}^\star}(\bar{f}_{\infty,\tilde{\lambda}^\star,x_0}), \psi_j \rangle_{\mathscr{H}_K} \langle Dl_{n,\tilde{\lambda}^\star}(\bar{f}_{\infty,\tilde{\lambda}^\star,x_0}), \psi_l \rangle_{\mathscr{H}_K} \}$$

$$\times \int \psi_j(x_0)\psi_l(x_0) \, \pi(x_0) \, dx_0$$

$$= \frac{1}{4} \sum_{j=1}^{\infty} \frac{1}{\left( j^\beta + \tilde{\lambda}^\star j^k \right)^2} \mathbb{E}\{ \langle Dl_{n,\tilde{\lambda}^\star}(\bar{f}_{\infty,\tilde{\lambda}^\star,x_0}), \psi_j \rangle_{\mathscr{H}_K}^2 \}$$

To find the asymptotic order of $\|\tilde{f}_n - \bar{f}_n\|^2$, we need to derive the coefficients $\sum_{j=1}^{\infty} \frac{1}{\left( j^\beta + \tilde{\lambda}^\star j^k \right)^2}$ and $\mathbb{E}\{ < Dl_{n,\tilde{\lambda}^\star}(\bar{f}_{\infty,\tilde{\lambda}^\star,x_0}), \psi_j >_{\mathscr{H}_{\mathscr{K}}}^2 \}$.

**Derivation for** $\mathbb{E}\{ < Dl_{n\tilde{\lambda}^\star}(\bar{f}_{\infty,\tilde{\lambda}^\star,x_0}), \psi_j >_{\mathscr{H}_{\mathscr{K}}}^2 \}$.

$$\mathbb{E}\left\{ \left\langle Dl_{n,\tilde{\lambda}^\star}(\bar{f}_{\infty,\tilde{\lambda}^\star,x_0}), \psi_j \right\rangle_{\mathscr{H}_K}^2 \right\}$$

$$= \mathbb{E}\left\{ \left\langle Dl_n(\bar{f}_{\infty,\tilde{\lambda}^\star,x_0}) - Dl_\infty(\bar{f}_{\infty,\tilde{\lambda}^\star,x_0}), \, \psi_j \right\rangle_{\mathscr{H}_K} \right\}^2$$

$$= \mathbb{E}\left[ -\frac{2}{n} \sum_{i \in \mathscr{S}(x_0)} \{y_i - \bar{f}_{\infty,\tilde{\lambda}^\star,x_0}(X_i)\}\psi_j(X_i) + 2\sum_{i=1}^{N} \omega_i(x_0) \, \mathbb{E}\left\{ (f_0(X_i) - \bar{f}_{\infty,\tilde{\lambda}^\star,x_0}(X_i)\}\psi_j(X_i) \,\big|\, X_i \right\} \right]^2$$

$$= \mathbb{E}\left[ -\frac{2}{n} \sum_{i=1}^{N} I_i(x_0) \{y_i - \bar{f}_{\infty,\tilde{\lambda}^\star,x_0}(X_i)\}\psi_j(X_i) + \frac{2}{n} \mathbb{E}\left\{ \sum_{i=1}^{N} I_i(x_0) \left( f_0(X_i) - \bar{f}_{\infty,\tilde{\lambda}^\star,x_0}(X_i) \right)\psi_j(X_i) \,\big|\, X_i, y_i \right\} \right]^2$$

$$= \frac{4}{n^2} \mathbb{E}\left\{ \sum_{i=1}^{N} \left( I_i(x_0) - n\,\omega_i(x_0) \right) (y_i - \bar{f}_{\infty,\tilde{\lambda}^\star,x_0}(X_i))\psi_j(X_i) \right\}^2 \quad \text{(since } \mathbb{E}[I_i(x_0) \mid X_i, y_i] = n\omega_i(x_0))$$

$$= \frac{4}{n^2} \mathbb{E}\left[ \mathbb{E}\left\{ \left( \sum_{i=1}^{N} \left( I_i(x_0) - n\,\omega_i(x_0) \right) (y_i - \bar{f}_{\infty,\tilde{\lambda}^\star,x_0}(X_i))\psi_j(X_i) \right)^2 \,\big|\, X_i, y_i \right\} \right]$$

$$= \frac{4}{n^2} \mathbb{E}\left[ \text{Var}\left\{ \sum_{i=1}^{N} I_i(x_0)\,a_i \,\big|\, X_i, y_i \right\} + \left\{ \mathbb{E}\left\{ \sum_{i=1}^{N} I_i(x_0)\,a_i \,\big|\, X_i, y_i \right\} - n\sum_{i=1}^{N} \omega_i(x_0)a_i \right\}^2 \right]$$

$$= \frac{4}{n^2} \, \mathbb{E}\Big[ \mathrm{Var}\Big\{ \sum_{i=1}^{N} I_i(x_0) \, a_i \,\Big|\, X_i, y_i \Big\} \Big]$$

where we denoted $a_i := (y_i - \bar{f}_{\infty, \tilde{\lambda}^\star, x_0}(X_i))\psi_j(X_i)$. Using the multinomial covariance, $\mathrm{Var}(I_i \mid X_i) = n\omega_i(1 - \omega_i)$ and $\mathrm{Cov}(I_i, I_k \mid x) = -n\omega_i\omega_k$ for $i \neq k$, we get

$$\mathrm{Var}\Big\{ \sum_{i=1}^{N} I_i(x_0) \, a_i \,\Big|\, X_i, y_i \Big\} = \sum_{i=1}^{N} a_i^2 \, \mathrm{Var}(I_i \mid X_i) + 2 \sum_{1 \leq i < k \leq N} a_i a_k \, \mathrm{Cov}(I_i, I_k \mid x)$$

$$= n \sum_{i=1}^{N} \omega_i a_i^2 - n \sum_{i=1}^{N} \sum_{k=1}^{N} \omega_i \omega_k a_i a_k$$

$$= n\Big\{ \sum_{i=1}^{N} \omega_i a_i^2 - \Big( \sum_{i=1}^{N} \omega_i a_i \Big)^2 \Big\} \;\leq\; n \sum_{i=1}^{N} \omega_i a_i^2.$$

Therefore,

$$\mathbb{E}\Big\{ \Big\langle Dl_{n,\tilde{\lambda}^\star}(\bar{f}_{\infty, \tilde{\lambda}^\star, x_0}), \psi_j \Big\rangle_{\mathscr{H}_K}^2 \Big\}$$

$$\leq \frac{4}{n} \, \mathbb{E}\Big\{ \sum_{i=1}^{N} \omega_i(x_0) \, a_i^2 \Big\}$$

$$= \frac{4}{n} \sum_{i=1}^{N} \omega_i(x_0) \, \mathbb{E}\Big[ \{y_i - \bar{f}_{\infty, \tilde{\lambda}^\star, x_0}(X_i)\}^2 \, \psi_j(X_i)^2 \Big]$$

$$= \frac{4}{n} \sum_{i=1}^{N} \omega_i(x_0) \, \mathbb{E}\Big[ \big\{ (f_0(X_i) - \bar{f}_{\infty, \tilde{\lambda}^\star, x_0}(X_i)) + \epsilon_i \big\}^2 \, \psi_j(X_i)^2 \Big]$$

$$= \frac{4}{n} \sum_{i=1}^{N} \omega_i(x_0) \, \mathbb{E}\Big[ \big\{ (f_0(X_i) - \bar{f}_{\infty, \tilde{\lambda}^\star, x_0}(X_i))^2 + \sigma^2 \big\} \, \psi_j(X_i)^2 \Big]$$

$$\leq \frac{4}{n} \Big[ \sigma^2 \sum_{i=1}^{N} \omega_i(x_0) \, \mathbb{E}\{\psi_j(X_i)^2\} + \sum_{i=1}^{N} \omega_i(x_0) \, \mathbb{E}\Big[ \{f_0(X_i) - \bar{f}_{\infty, \tilde{\lambda}^\star, x_0}(X_i)\}^2 \, \psi_j(X_i)^2 \Big] \Big] \asymp \frac{1}{n}.$$

**Derivation for the coefficients** $\sum_{j=1}^{\infty} \big(j^\beta + \tilde{\lambda}^\star j^k\big)^{-2}$. Assumed $\lambda_j \asymp j^{-k}$ with $k > 1$, and $\omega_{x_0, j} \asymp j^\beta$ uniformly in $x_0$ for some $\beta < k$ (so that $k - \beta > 0$). Consider

$$S_1(x_0) := \sum_{j=1}^{\infty} \frac{1}{\big(j^\beta + \tilde{\lambda}^\star j^k\big)^2}.$$

Let $J$ solve $j^\beta = \tilde{\lambda}^\star j^k$, i.e. $J \asymp (\tilde{\lambda}^\star)^{-1/(k-\beta)}$.

We split

$$S_1(x_0) = \sum_{j \leq J} \frac{1}{\big(j^\beta + \tilde{\lambda}^\star j^k\big)^2} \;+\; \sum_{j > J} \frac{1}{\big(j^\beta + \tilde{\lambda}^\star j^k\big)^2}.$$

For $j \leq J$, $j^\beta \gg \tilde{\lambda}^\star j^k$ and $(j^\beta + \tilde{\lambda}^\star j^k)^2 \asymp j^{2\beta}$, giving

$$\sum_{j \leq J} \frac{1}{\big(j^\beta + \tilde{\lambda}^\star j^k\big)^2} \;\asymp\; \sum_{j \leq J} j^{-2\beta} \;\asymp\; \begin{cases} J^{1-2\beta}, & \beta < \frac{1}{2}, \\ \log J, & \beta = \frac{1}{2}, \\ 1, & \frac{1}{2} < \beta < k. \end{cases} \qquad (4)$$

For $j > J$, $\tilde{\lambda}^\star j^k \gg j^\beta$ and $(j^\beta + \tilde{\lambda}^\star j^k)^2 \asymp (\tilde{\lambda}^\star)^2 j^{2k}$, giving

$$\sum_{j > J} \frac{1}{\big(j^\beta + \tilde{\lambda}^\star j^k\big)^2} \;\asymp\; \frac{1}{(\tilde{\lambda}^\star)^2} \sum_{j > J} j^{-2k} \;\asymp\; \frac{1}{(\tilde{\lambda}^\star)^2} \, J^{-(2k-1)} \;\asymp\; (\tilde{\lambda}^\star)^{\frac{2\beta-1}{k-\beta}}. \qquad (5)$$

Using $J \asymp (\tilde{\lambda}^\star)^{-1/(k-\beta)}$ to rewrite equation 4 in terms of $\tilde{\lambda}^\star$,

$$\sum_{j \le J} j^{-2\beta} \asymp \begin{cases} (\tilde{\lambda}^\star)^{-\frac{1-2\beta}{k-\beta}}, & \beta < \frac{1}{2}, \\ \log(1/\tilde{\lambda}^\star), & \beta = \frac{1}{2}, \\ 1, & \frac{1}{2} < \beta < k. \end{cases}$$

Comparing with equation 5, we obtain

$$S_1(x_0) \asymp \begin{cases} (\tilde{\lambda}^\star)^{-\frac{1-2\beta}{k-\beta}}, & \beta < \frac{1}{2}, \\ \log\left(1/\tilde{\lambda}^\star\right), & \beta = \frac{1}{2}, \\ 1, & \frac{1}{2} < \beta < k. \end{cases}$$

Since $\mathbb{E}[\langle Dl_{n,\tilde{\lambda}^\star}(\bar{f}_{\infty,\tilde{\lambda}^\star,x_0}), \psi_j \rangle_{\mathscr{H}_K}^2] \asymp n^{-1}$, we obtain

$$\|\tilde{f}_{n,\tilde{\lambda}^\star} - \bar{f}_{n,\tilde{\lambda}^\star}\|^2 = \begin{cases} O_p\left(n^{-1} (\tilde{\lambda}^\star)^{-\frac{1-2\beta}{k-\beta}}\right), & \beta < \frac{1}{2}, \\ O_p(n^{-1} \log(1/\tilde{\lambda}^\star)), & \beta = \frac{1}{2}, \\ O_p(n^{-1}), & \frac{1}{2} < \beta < k. \end{cases} \tag{A}$$

### B.1.2  STEP 2: EVALUATE THE ORDER OF $\hat{f}_{n,\tilde{\lambda}^\star} - \tilde{f}_{n,\tilde{\lambda}^\star}$.

For brevity, write

$$\hat{f}_{x_0} := \hat{f}_{n,\tilde{\lambda}^\star,x_0}, \qquad \bar{f}_{x_0} := \bar{f}_{\infty,\tilde{\lambda}^\star,x_0}, \qquad \tilde{f}_{x_0} := \tilde{f}_{n,\tilde{\lambda}^\star,x_0}.$$

By definition of $\hat{f}_{x_0}$, we know $Dl_{n,\tilde{\lambda}^\star}(\hat{f}_{x_0}) = 0$. We now check the following equation:

$$Dl_{n,\tilde{\lambda}^\star}(\hat{f}_{x_0}) = Dl_{n,\tilde{\lambda}^\star}(\bar{f}_{x_0}) + D^2 l_{n,\tilde{\lambda}^\star}(\bar{f}_{x_0})(\hat{f}_{x_0} - \bar{f}_{x_0}) = 0. \tag{6}$$

Using the functional derivatives derived above, we have

$$Dl_{n,\tilde{\lambda}^\star} = -\frac{2}{n} \sum_{i \in \mathscr{S}(x_0)} K(X_i, \cdot) \left(y_i - < \bar{f}_{x_0}, K(X_i, \cdot) >_{\mathscr{H}_K}\right) + 2\tilde{\lambda}^\star < \bar{f}_{x_0}, \cdot >_{\mathscr{H}_K} \tag{7}$$

$$D^2 l_{n,\tilde{\lambda}^\star}(\bar{f}_{x_0})(\hat{f}_{x_0} - \bar{f}_{x_0}) = \frac{2}{n} \sum_{i \in \mathscr{S}(x_0)} < \hat{f}_{x_0} - \bar{f}_{x_0}, K(X_i, \cdot) >_{\mathscr{H}_K} K(X_i, \cdot) + 2\tilde{\lambda}^\star < \hat{f}_{x_0} - \bar{f}_{x_0}, \cdot >_{\mathscr{H}_K} \tag{8}$$

Adding (7) and (8) together, we obtain:

$$\begin{aligned} (7) + (8) &= -\frac{2}{n} \sum_{i \in \mathscr{S}(x_0)} K(X_i, \cdot)(y_i - < \hat{f}_{x_0}, K(X_i, \cdot) >_{\mathscr{H}_K}) + 2\tilde{\lambda}^\star < \hat{f}_{x_0}, \cdot >_{\mathscr{H}_K} \\ &= Dl_{n,\tilde{\lambda}^\star}(\hat{f}_{x_0}) = 0. \end{aligned}$$

Using the definition of $\tilde{f}_{x_0}$, we have

$$Dl_{n,\tilde{\lambda}^\star}(\bar{f}_{x_0}) + D^2 l_{\infty,\tilde{\lambda}^\star}(\bar{f}_{x_0})(\tilde{f}_{x_0} - \bar{f}_{x_0}) = 0.$$

Combining with the equation $Dl_{n,\tilde{\lambda}^\star}(\hat{f}_{x_0}) = Dl_{n,\tilde{\lambda}^\star}(\bar{f}_{x_0}) + D^2 l_{n,\tilde{\lambda}^\star}(\bar{f}_{x_0})(\hat{f}_{x_0} - \bar{f}_{x_0}) = 0$, we get

$$D^2 l_{\infty,\tilde{\lambda}^\star}(\bar{f}_{x_0})(\tilde{f}_{x_0} - \bar{f}_{x_0}) = D^2 l_{n,\tilde{\lambda}^\star}(\bar{f}_{x_0})(\hat{f}_{x_0} - \bar{f}_{x_0}).$$

Then, we can derive:

$$D^2 l_{\infty,\tilde{\lambda}^\star}(\bar{f}_{x_0})(\hat{f}_{x_0} - \tilde{f}_{x_0}) = D^2 l_{\infty,\tilde{\lambda}^\star}(\bar{f}_{x_0})(\hat{f}_{x_0} - \bar{f}_{x_0}) + D^2 l_{\infty,\tilde{\lambda}^\star}(\bar{f}_{x_0})(\bar{f}_{x_0} - \tilde{f}_{x_0})$$

$$= D^2 l_{\infty,\tilde{\lambda}^\star}(\bar{f}_{x_0})(\hat{f}_{x_0} - \bar{f}_{x_0}) - D^2 l_{n,\tilde{\lambda}^\star}(\bar{f}_{x_0})(\hat{f}_{x_0} - \bar{f}_{x_0})$$

$$= D^2 l_\infty(\bar{f}_{x_0})(\hat{f}_{x_0} - \bar{f}_{x_0}) - D^2 l_n(\bar{f}_{x_0})(\hat{f}_{x_0} - \bar{f}_{x_0}),$$

where in the last line we used that the penalty part is the same in $D^2 l_{\infty,\tilde{\lambda}^\star}$ and $D^2 l_{n,\tilde{\lambda}^\star}$.

Then $\hat{f}_{x_0} - \tilde{f}_{x_0}$ can be expressed as

$$\hat{f}_{x_0} - \tilde{f}_{x_0} = \frac{1}{2} G_{\tilde{\lambda}^\star}^{-1} \Big\{ D^2 l_\infty(\bar{f}_{x_0})(\hat{f}_{x_0} - \bar{f}_{x_0}) - D^2 l_n(\bar{f}_{x_0})(\hat{f}_{x_0} - \bar{f}_{x_0}) \Big\}.$$

So we write:

$$\|\hat{f}_{n,\tilde{\lambda}^\star} - \tilde{f}_{n,\tilde{\lambda}^\star}\|^2$$

$$:= \int \big\langle \hat{f}_{n,\tilde{\lambda}^\star,x_0} - \tilde{f}_{n,\tilde{\lambda}^\star,x_0}, K(x_0,\cdot) \big\rangle_{\mathscr{H}_K}^2 \pi(x_0)\, dx_0$$

$$= \int \left( \sum_{j=1}^\infty \lambda_j \big\langle \hat{f}_{n,\tilde{\lambda}^\star,x_0} - \tilde{f}_{n,\tilde{\lambda}^\star,x_0}, \psi_j \big\rangle_{\mathscr{H}_K} \psi_j(x_0) \right)^2 \pi(x_0)\, dx_0$$

$$= \frac{1}{4} \int \left\{ \sum_{j=1}^\infty \lambda_j \Big\langle G_{\tilde{\lambda}^\star}^{-1} \Big( D^2 l_{\infty,\tilde{\lambda}^\star}(\bar{f}_{x_0}) - D^2 l_{n,\tilde{\lambda}^\star}(\bar{f}_{x_0}) \Big) [\hat{f}_{x_0} - \bar{f}_{x_0}], \psi_j \Big\rangle_{\mathscr{H}_K} \psi_j(x_0) \right\}^2 \pi(x_0)\, dx_0$$

$$= \frac{1}{4} \int \left[ \sum_{j=1}^\infty \big(\omega_{x_0,j} + \tilde{\lambda}^\star \lambda_j^{-1}\big)^{-1} \left\{ \begin{array}{l} \big\langle D^2 l_{\infty,\tilde{\lambda}^\star}(\bar{f}_{x_0})(\hat{f}_{x_0} - \bar{f}_{x_0}), \psi_j \big\rangle_{\mathscr{H}_K} \\ \quad - \big\langle D^2 l_{n,\tilde{\lambda}^\star}(\bar{f}_{x_0})(\hat{f}_{x_0} - \bar{f}_{x_0}), \psi_j \big\rangle_{\mathscr{H}_K} \end{array} \right\} \psi_j(x_0) \right]^2 \pi(x_0)\, dx_0$$

$$=: \frac{1}{4} \int \left\{ \sum_{j=1}^\infty \big(\omega_{x_0,j} + \tilde{\lambda}^\star \lambda_j^{-1}\big)^{-1} A_j(x_0)\, \psi_j(x_0) \right\}^2 \pi(x_0)\, dx_0, \tag{2.1}$$

where

$$A_j(x_0) = \big\langle D^2 l_{\infty,\tilde{\lambda}^\star}(\bar{f}_{x_0})(\hat{f}_{x_0} - \bar{f}_{x_0}), \psi_j \big\rangle_{\mathscr{H}_K} - \big\langle D^2 l_{n,\tilde{\lambda}^\star}(\bar{f}_{x_0})(\hat{f}_{x_0} - \bar{f}_{x_0}), \psi_j \big\rangle_{\mathscr{H}_K}$$

$$= 2 \sum_{\ell=1}^\infty \lambda_\ell^2 \big\langle \hat{f}_{x_0} - \bar{f}_{x_0}, \psi_\ell \big\rangle_{\mathscr{H}_K} \big\langle \psi_j, \psi_\ell \big\rangle_{\mathscr{H}_K} \mathbb{E}\Big\{ \sum_{i=1}^N \omega_i(x_0)\, \psi_\ell^2(X_i) \Big\}$$

$$\quad - \frac{2}{n} \sum_{i \in \mathscr{S}(x_0)} \sum_{k=1}^\infty \lambda_k^2 \big\langle \hat{f}_{x_0} - \bar{f}_{x_0}, \psi_k \big\rangle_{\mathscr{H}_K} \big\langle \psi_j, \psi_k \big\rangle_{\mathscr{H}_K} \psi_k^2(X_i)$$

$$= 2 \lambda_j^2 \frac{1}{\lambda_j} \big\langle \hat{f}_{x_0} - \bar{f}_{x_0}, \psi_j \big\rangle_{\mathscr{H}_K} \mathbb{E}\Big\{ \sum_{i=1}^N \omega_i(x_0)\, \psi_j^2(X_i) \Big\}$$

$$\quad - \frac{2}{n} \sum_{i \in \mathscr{S}(x_0)} \lambda_j^2 \frac{1}{\lambda_j} \big\langle \hat{f}_{x_0} - \bar{f}_{x_0}, \psi_j \big\rangle_{\mathscr{H}_K} \psi_j^2(X_i)$$

$$= 2 \lambda_j \big\langle \hat{f}_{x_0} - \bar{f}_{x_0}, \psi_j \big\rangle_{\mathscr{H}_K} \left[ \mathbb{E}\Big\{ \sum_{i=1}^N \omega_i(x_0)\, \psi_j^2(X_i) \Big\} - \frac{1}{n} \sum_{i \in \mathscr{S}(x_0)} \psi_j^2(X_i) \right].$$

plug it back into 2.1

$$= \frac{1}{4} \int \left[ \sum_{j=1}^\infty \big(\omega_{x_0,j} + \tilde{\lambda}^\star \lambda_j^{-1}\big)^{-1} \left\{ \Big( 2 \lambda_j \big\langle \hat{f}_{x_0} - \bar{f}_{x_0}, \psi_j \big\rangle_{\mathscr{H}_K} \Big) \Big( \omega_{x_0,j} - \tfrac{1}{n} \sum_{i \in \mathscr{S}(x_0)} \psi_j^2(X_i) \Big) \psi_j(x_0) \right\} \right]^2 \pi(x_0)\, dx_0$$

$$= \frac{1}{n^2} \int \left[ \sum_{j=1}^{\infty} \lambda_j \left( \omega_{x_0,j} + \tilde{\lambda}^{\star} \lambda_j^{-1} \right)^{-1} \left\langle \hat{f}_{x_0} - \bar{f}_{x_0}, \psi_j \right\rangle_{\mathscr{H}_K} \psi_j(x_0) \left\{ n \, \omega_{x_0,j} - \sum_{i \in \mathscr{S}(x_0)} \psi_j^2(X_i) \right\} \right]^2 \pi(x_0) \, dx_0$$

$$= \frac{1}{n^2} \int \left[ \sum_{j=1}^{\infty} \lambda_j \left( \omega_{x_0,j} + \tilde{\lambda}^{\star} \lambda_j^{-1} \right)^{-1} \left\langle \hat{f}_{x_0} - \bar{f}_{x_0}, \psi_j \right\rangle_{\mathscr{H}_K} \psi_j(x_0) \right.$$

$$\left. \times \left\{ n \mathbb{E} \left\{ \sum_{i=1}^{N} \omega_i(x_0) \psi_j^2(X_i) \right\} - \sum_{i=1}^{N} I_i(x_0) \, \psi_j^2(X_i) \right\} \right]^2 \pi(x_0) \, dx_0. \tag{2.2}$$

Set

$$\Delta_j(x_0) := n \, \mathbb{E} \left\{ \sum_{i=1}^{N} \omega_i(x_0) \, \psi_j^2(X_i) \right\} - \sum_{i=1}^{N} I_i(x_0) \, \psi_j^2(X_i).$$

Known that $\mathbb{E}\{I_i(x_0) \mid X_1, \ldots, X_N\} = n \, \omega_i(x_0)$ and $\sum_{i=1}^{N} \omega_i(x_0) = 1$, $\mathbb{E}\{\Delta_j(x_0)\} = 0$. For the second moment, condition on $\{X_i\}_{i=1}^{N}$, then

$$\mathbb{E}\{\Delta_j(x_0)^2\} = \mathbb{E}\left[ \text{Var}\left\{ \sum_{i=1}^{N} I_i(x_0) \, \psi_j^2(X_i) \,\Big|\, X_1, \ldots, X_N \right\} \right]$$

$$= \mathbb{E}\left[ n \left\{ \sum_{i=1}^{N} \omega_i(x_0) \, \psi_j^4(X_i) - \left( \sum_{i=1}^{N} \omega_i(x_0) \, a_i \right)^2 \right\} \right]$$

$$\leq \mathbb{E}\left\{ n \sum_{i=1}^{N} \omega_i(x_0) \, \psi_j^4(X_i) \right\}$$

$$= n \, \mathbb{E}\left\{ \sum_{i=1}^{N} \omega_i(x_0) \, \psi_j^4(X_i) \right\} = n \, \mathbb{E}\{\psi_j^4(X)\}.$$

Therefore,

$$\Delta_j(x_0) \asymp O_p\left( \sqrt{n \, \mathbb{E}\{\psi_j^4(X)\}} \right).$$

Plug it back to (2.2):

$$\asymp \frac{1}{n^2} \int \left\{ \sum_{j=1}^{\infty} \lambda_j \left( \omega_{x_0,j} + \tilde{\lambda}^{\star} \lambda_j^{-1} \right)^{-1} \left\langle \hat{f}_{x_0} - \bar{f}_{x_0}, \psi_j \right\rangle_{\mathscr{H}_K} \psi_j(x_0) \right\}^2 \pi(x_0) \, dx_0$$

$$= \frac{1}{n^2} \int \left[ \sum_{j=1}^{\infty} (\omega_{x_0,j} + \tilde{\lambda}^{\star} \lambda_j^{-1})^{-1} \left\{ \lambda_j \left\langle \hat{f}_{x_0} - \bar{f}_{x_0}, \psi_j \right\rangle_{\mathscr{H}_K} \psi_j(x_0) \right\} \right]^2 \pi(x_0) \, dx_0$$

$$\leq \frac{1}{n^2} \int \left\{ \sum_{j=1}^{\infty} (\omega_{x_0,j} + \tilde{\lambda}^{\star} \lambda_j^{-1})^{-2} \right\} \left\{ \sum_{j=1}^{\infty} \lambda_j^2 \left\langle \hat{f}_{x_0} - \bar{f}_{x_0}, \psi_j \right\rangle_{\mathscr{H}_K}^2 \psi_j(x_0)^2 \right\} \pi(x_0) \, dx_0$$

$$\leq \frac{1}{n^2} \int \underbrace{\left\{ \sum_{j=1}^{\infty} (\omega_{x_0,j} + \tilde{\lambda}^{\star} \lambda_j^{-1})^{-2} \right\}}_{S_1(x_0)} \left\{ \sum_{j=1}^{\infty} \lambda_j \left\langle \hat{f}_{x_0} - \bar{f}_{x_0}, \psi_j \right\rangle_{\mathscr{H}_K} \psi_j(x_0) \right\}^2 \pi(x_0) \, dx_0.$$

Assume $\lambda_j \asymp j^{-k}$ with $k > 1$ and, uniformly in $x_0$, $\omega_{x_0,j} \asymp j^\beta$ for some $\beta < k$. Recall

$$S_1(x_0) := \sum_{j=1}^{\infty} \left(\omega_{x_0,j} + \tilde{\lambda}^\star \lambda_j^{-1}\right)^{-2} \asymp \sum_{j=1}^{\infty} \frac{1}{\left(j^\beta + \tilde{\lambda}^\star j^k\right)^2} \asymp \begin{cases} (\tilde{\lambda}^\star)^{-\frac{1-2\beta}{k-\beta}}, & \beta < \frac{1}{2}, \\[2mm] \log(1/\tilde{\lambda}^\star), & \beta = \frac{1}{2}, \\[2mm] 1, & \frac{1}{2} < \beta < k. \end{cases}$$

Consequently,

$$\|\hat{f}_{n,\tilde{\lambda}^\star} - \tilde{f}_{n,\tilde{\lambda}^\star}\|^2 \leq \frac{1}{n^2} \int \underbrace{\sum_{j=1}^{\infty} \left(\omega_{x_0,j} + \tilde{\lambda}^\star \lambda_j^{-1}\right)^{-2}}_{= S_1(x_0)} \left\{ \sum_{j=1}^{\infty} \lambda_j \langle \hat{f}_{x_0} - \bar{f}_{x_0}, \psi_j \rangle_{\mathscr{H}_K} \psi_j(x_0) \right\}^2 \pi(x_0)\, dx_0$$

$$= \frac{S_1(x_0)}{n^2} \sum_{j=1}^{\infty} \lambda_j^2 \langle \hat{f}_{n,\tilde{\lambda}^\star} - \bar{f}_{n,\tilde{\lambda}^\star}, \psi_j \rangle_{\mathscr{H}_K}^2$$

$$= \frac{S_1(x_0)}{n^2} \|\hat{f}_{n,\tilde{\lambda}^\star} - \bar{f}_{n,\tilde{\lambda}^\star}\|^2,$$

with $S_1(x_0)$ as above. Therefore,

$$\|\hat{f}_{n,\tilde{\lambda}^\star} - \tilde{f}_{n,\tilde{\lambda}^\star}\|^2 = \begin{cases} O_p\!\left(n^{-2}\,(\tilde{\lambda}^\star)^{-\frac{1-2\beta}{k-\beta}} \|\hat{f}_{n,\tilde{\lambda}^\star} - \bar{f}_{n,\tilde{\lambda}^\star}\|^2\right), & \beta < \frac{1}{2}, \\[2mm] O_p\!\left(n^{-2}\,\log(1/\tilde{\lambda}^\star) \|\hat{f}_{n,\tilde{\lambda}^\star} - \bar{f}_{n,\tilde{\lambda}^\star}\|^2\right), & \beta = \frac{1}{2}, \\[2mm] O_p\!\left(n^{-2} \|\hat{f}_{n,\tilde{\lambda}^\star} - \bar{f}_{n,\tilde{\lambda}^\star}\|^2\right), & \frac{1}{2} < \beta < k. \end{cases} \tag{B}$$

Recall

$$\|\tilde{f}_{n,\tilde{\lambda}^\star} - \bar{f}_{n,\tilde{\lambda}^\star}\|^2 = \begin{cases} O_p\!\left(n^{-1}\,(\tilde{\lambda}^\star)^{-\frac{1-2\beta}{k-\beta}}\right), & \beta < \frac{1}{2}, \\[2mm] O_p(n^{-1}\log(1/\tilde{\lambda}^\star)), & \beta = \frac{1}{2}, \\[2mm] O_p(n^{-1}), & \frac{1}{2} < \beta < k. \end{cases} \tag{A}$$

From (B), write

$$\|\hat{f}_{n,\tilde{\lambda}^\star} - \tilde{f}_{n,\tilde{\lambda}^\star}\|^2 = \eta_n(\tilde{\lambda}^\star, \beta) \|\hat{f}_{n,\tilde{\lambda}^\star} - \bar{f}_{n,\tilde{\lambda}^\star}\|^2,$$

where

$$\eta_n(\tilde{\lambda}^\star, \beta) = \begin{cases} O_p(n^{-2}(\tilde{\lambda}^\star)^{-\frac{1-2\beta}{k-\beta}}), & \beta < \frac{1}{2}, \\[2mm] O_p(n^{-2}\log(1/\tilde{\lambda}^\star)), & \beta = \frac{1}{2}, \\[2mm] O_p(n^{-2}), & \frac{1}{2} < \beta < k. \end{cases}$$

By the triangle inequality,

$$\|\hat{f}_{n,\tilde{\lambda}^\star} - \bar{f}_{n,\tilde{\lambda}^\star}\|^2 \leq \|\tilde{f}_{n,\tilde{\lambda}^\star} - \bar{f}_{n,\tilde{\lambda}^\star}\|^2 + \|\hat{f}_{n,\tilde{\lambda}^\star} - \tilde{f}_{n,\tilde{\lambda}^\star}\|^2. \tag{C}$$

If $\eta_n(\tilde{\lambda}^\star, \beta) \xrightarrow{p} 0$, then

$$\begin{cases} n^{-2}(\tilde{\lambda}^\star)^{-\frac{1-2\beta}{k-\beta}} \to 0, & \beta < \frac{1}{2}, \\[2mm] n^{-2}\log(1/\tilde{\lambda}^\star) \to 0, & \beta = \frac{1}{2}, \\[2mm] n^{-2} \to 0, & \frac{1}{2} < \beta < k, \end{cases} \tag{D}$$

then from (C) we get

$$(1-\eta_n)\,\|\hat{f}_{n,\tilde{\lambda}^\star}-\bar{f}_{n,\tilde{\lambda}^\star}\|^2 \;\leq\; \|\tilde{f}_{n,\tilde{\lambda}^\star}-\bar{f}_{n,\tilde{\lambda}^\star}\|^2 \quad\text{and}\quad \|\tilde{f}_{n,\tilde{\lambda}^\star}-\bar{f}_{n,\tilde{\lambda}^\star}\|^2 \;\geq\; (1-\eta_n)\,\|\hat{f}_{n,\tilde{\lambda}^\star}-\bar{f}_{n,\tilde{\lambda}^\star}\|^2,$$

hence, under (D),

$$\|\hat{f}_{n,\tilde{\lambda}^\star} - \bar{f}_{n,\tilde{\lambda}^\star}\|^2 = \{1 + o_p(1)\}\,\|\tilde{f}_{n,\tilde{\lambda}^\star} - \bar{f}_{n,\tilde{\lambda}^\star}\|^2.$$

Combining with (A) yields

$$\|\hat{f}_{n,\tilde{\lambda}^\star} - \bar{f}_{n,\tilde{\lambda}^\star}\|^2 \;=\; \begin{cases} O_p\!\Big(n^{-1}\,(\tilde{\lambda}^\star)^{-\frac{1-2\beta}{k-\beta}}\Big), & \beta < \dfrac{1}{2}, \\[2ex] O_p\big(n^{-1}\log(1/\tilde{\lambda}^\star)\big), & \beta = \dfrac{1}{2}, \\[2ex] O_p(n^{-1}), & \dfrac{1}{2} < \beta < k, \end{cases} \tag{E}$$

when (D) holds.

**Condition (D) and corresponding constraints.**

For $\eta_n(\tilde{\lambda}^\star,\beta) \to 0$ in the case $\omega_{x_0,j} \asymp j^\beta$, the corresponding rate constraints on $\tilde{\lambda}^\star$ are:

- *Case $\beta < \frac{1}{2}$:* We require

$$n^{-2}\,(\tilde{\lambda}^\star)^{-\frac{1-2\beta}{k-\beta}} \to 0,$$

  equivalently,

$$(\tilde{\lambda}^\star)^{\frac{1-2\beta}{k-\beta}} \gg n^{-2} \quad\Longleftrightarrow\quad \tilde{\lambda}^\star \gg n^{-\frac{2(k-\beta)}{1-2\beta}}.$$

- *Case $\beta = \frac{1}{2}$:* We require

$$n^{-2}\log\big(1/\tilde{\lambda}^\star\big) \to 0,$$

  equivalently,

$$\log\big(1/\tilde{\lambda}^\star\big) = o(n^2).$$

- *Case $\frac{1}{2} < \beta < k$:* Here $n^{-2} \to 0$ automatically; no further condition on $\tilde{\lambda}^\star$ is needed.

### B.2 ASYMPTOTIC ORDER FOR DETERMINISTIC ERROR.

Recall

$$l_{\infty,x_0}(f) := \mathbb{E}\big\{l_{n,x_0}(f)\big\} = \sigma^2 \;+\; \mathbb{E}\bigg\{\sum_{i=1}^{N}\omega_i(x_0)\,\big\langle f - f_0,\,K(X_i,\cdot)\big\rangle_{\mathscr{H}_K}^2\bigg\}$$

$$= \sigma^2 + \sum_{i=1}^{N}\mathbb{E}\Big[\omega_i(x_0)\big\{f(X_i) - f_0(X_i)\big\}^2\Big]$$

$$= \sigma^2 + \sum_{i=1}^{N}\mathbb{E}\bigg[\omega_i(x_0)\Big\{\sum_{j=1}^{\infty}(c_j - a_j)\psi_j(X_i)\Big\}^2\bigg]$$

$$= \sigma^2 + \sum_{j,\ell\geq 1}(c_j - a_j)(c_\ell - a_\ell)\sum_{i=1}^{N}\mathbb{E}\{\omega_i(x_0)\psi_j(X_i)\psi_\ell(X_i)\}$$

$$= \sigma^2 + \sum_{j=1}^{\infty}\omega_{x_0,j}(c_j - a_j)^2,$$

where $\omega_{x_0,j} := \mathbb{E}\big\{\sum_{i=1}^{N}\omega_i(x_0)\psi_j^2(X_i)\big\}$.

The objective function about $c_j$ at $x_0$ becomes

$$Q_{x_0}(c) = \sum_{j\geq 1}\Big\{\omega_{x_0,j}\,(c_j - a_j)^2 + \tilde{\lambda}^\star\,\frac{c_j^2}{\lambda_j}\Big\}.$$

Taking derivatives,

$$2\omega_{x_0,j}(c_j - a_j) + 2\tilde{\lambda}^{\star}\frac{c_j}{\lambda_j} = 0 \implies (\omega_{x_0,j} + \tilde{\lambda}^{\star}\lambda_j^{-1})\,c_j = \omega_{x_0,j}a_j,$$

hence

$$\bar{c}_j(x_0) = \frac{\omega_{x_0,j}}{\omega_{x_0,j} + \tilde{\lambda}^{\star}\lambda_j^{-1}}\,a_j \quad \text{and} \quad \bar{c}_j(x_0) - a_j = -\frac{\tilde{\lambda}^{\star}}{\omega_{x_0,j}\lambda_j + \tilde{\lambda}^{\star}}\,a_j.$$

Assume $\lambda_j \asymp j^{-k}$ with $k > 1$, $a_j^2\lambda_j^{-1} = j^{-a}$ with $a > 1$ (so $a_j^2 \asymp j^{-(a+k)}$), and fix $\beta < k$ such that, uniformly in $x_0$,

$$\omega_{x_0,j} \asymp j^{\beta} \qquad \text{(equivalently, } \lambda_j\omega_{x_0,j} \asymp j^{\beta-k}\text{)}.$$

Now we evaluate $\|\bar{f}_{n,\tilde{\lambda}^{\star}} - f_0\|^2$:

$$\|\bar{f}_{n,\tilde{\lambda}^{\star}} - f_0\|^2 := \int \left\langle \bar{f}_{n,\tilde{\lambda}^{\star},x_0} - f_0,\, K(x_0,\cdot) \right\rangle_{\mathscr{H}_K}^2 \pi(x_0)\,dx_0$$

$$= \int \left\{\sum_{j=1}^{\infty}\lambda_j \left\langle \bar{f}_{n,\tilde{\lambda}^{\star},x_0} - f_0, \psi_j\right\rangle_{\mathscr{H}_K}\psi_j(x_0)\right\}^2 \pi(x_0)\,dx_0$$

$$= \int \left[\sum_{j=1}^{\infty}\left\{\bar{c}_j(x_0) - a_j\right\}\psi_j(x_0)\right]^2 \pi(x_0)\,dx_0$$

$$= \int \left\{\sum_{j=1}^{\infty}\frac{\tilde{\lambda}^{\star}}{\omega_{x_0,j}\lambda_j + \tilde{\lambda}^{\star}}\,a_j\,\psi_j(x_0)\right\}^2 \pi(x_0)\,dx_0$$

$$\asymp \int \left\{\sum_{j=1}^{\infty}\frac{\tilde{\lambda}^{\star}}{j^{\beta-k} + \tilde{\lambda}^{\star}}\,a_j\,\psi_j(x_0)\right\}^2 \pi(x_0)\,dx_0$$

$$= \sum_{j=1}^{\infty}\left(\frac{\tilde{\lambda}^{\star}}{j^{\beta-k} + \tilde{\lambda}^{\star}}\,a_j\right)^2 \asymp \sum_{j=1}^{\infty}\frac{(\tilde{\lambda}^{\star})^2\,j^{-(a+k)}}{\left(j^{\beta-k} + \tilde{\lambda}^{\star}\right)^2}.$$

Let $J \asymp (\tilde{\lambda}^{\star})^{-1/(k-\beta)}$ so that $J^{\beta-k} \asymp \tilde{\lambda}^{\star}$. Split at $J$:

$$\sum_{j\le J}\frac{(\tilde{\lambda}^{\star})^2\,j^{-(a+k)}}{\left(j^{\beta-k}\right)^2} \asymp (\tilde{\lambda}^{\star})^2\sum_{j\le J}j^{-(a+2\beta-k)}, \qquad \sum_{j>J}\frac{(\tilde{\lambda}^{\star})^2\,j^{-(a+k)}}{(\tilde{\lambda}^{\star})^2} = \sum_{j>J}j^{-(a+k)}.$$

Using the integral test directly on the exponent $a + 2\beta - k$ (with $a + k > 1$), we have

$$\sum_{j\le J}j^{-(a+2\beta-k)} \asymp \begin{cases} J^{1-(a+2\beta-k)}, & a + 2\beta - k < 1, \\ \log J, & a + 2\beta - k = 1, \\ 1, & a + 2\beta - k > 1, \end{cases} \qquad \sum_{j>J}j^{-(a+k)} \asymp J^{-(a+k-1)}.$$

With $J \asymp (\tilde{\lambda}^{\star})^{-1/(k-\beta)}$, this gives

$$(\tilde{\lambda}^{\star})^2\sum_{j\le J}j^{-(a+2\beta-k)} \asymp \begin{cases} (\tilde{\lambda}^{\star})^{\frac{a+k-1}{k-\beta}}, & a + 2\beta - k < 1, \\ (\tilde{\lambda}^{\star})^2\,\log(1/\tilde{\lambda}^{\star}), & a + 2\beta - k = 1, \\ (\tilde{\lambda}^{\star})^2, & a + 2\beta - k > 1, \end{cases} \qquad \sum_{j>J}j^{-(a+k)} \asymp (\tilde{\lambda}^{\star})^{\frac{a+k-1}{k-\beta}}.$$

Comparing the two parts yields, uniformly in $x_0$ and for any $\beta < k$,

$$\|\bar{f}_{n,\tilde{\lambda}^{\star}} - f_0\|^2 \asymp \begin{cases} (\tilde{\lambda}^{\star})^{\frac{a+k-1}{k-\beta}}, & a + 2\beta - k < 1, \\ (\tilde{\lambda}^{\star})^2\,\log(1/\tilde{\lambda}^{\star}), & a + 2\beta - k = 1, \\ (\tilde{\lambda}^{\star})^2, & a + 2\beta - k > 1. \end{cases} \tag{F}$$

### B.3 ASYMPTOTIC ORDER FOR MSE AND TUNING.

Assume $a > 1$, $k > 1$, and $\beta < k$.

**Deterministic error (F) (split by $a + 2\beta - k$).**

$$\|\bar{f}_{n,\tilde{\lambda}^\star} - f_0\|^2 \asymp \begin{cases} (\tilde{\lambda}^\star)^{\frac{a+k-1}{k-\beta}}, & a + 2\beta - k < 1, \\ (\tilde{\lambda}^\star)^2 \log(1/\tilde{\lambda}^\star), & a + 2\beta - k = 1, \\ (\tilde{\lambda}^\star)^2, & a + 2\beta - k > 1. \end{cases}$$

**Stochastic error (E) (split by $\beta = \frac{1}{2}$).** When condition (D) holds and $\beta < k$,

$$\|\hat{f}_{n,\tilde{\lambda}^\star} - \bar{f}_{n,\tilde{\lambda}^\star}\|^2 = \begin{cases} O_p\left(n^{-1}(\tilde{\lambda}^\star)^{-\frac{1-2\beta}{k-\beta}}\right), & \beta < \frac{1}{2}, \\ O_p\left(n^{-1}\log(1/\tilde{\lambda}^\star)\right), & \beta = \frac{1}{2}, \\ O_p(n^{-1}), & \frac{1}{2} < \beta < k. \end{cases}$$

**Condition (D).** We require $\eta_n(\tilde{\lambda}^\star, \beta) \to 0$:

$$\begin{cases} n^{-2}(\tilde{\lambda}^\star)^{-\frac{1-2\beta}{k-\beta}} \to 0, & \beta < \frac{1}{2}, \\ n^{-2}\log(1/\tilde{\lambda}^\star) \to 0, & \beta = \frac{1}{2}, \\ n^{-2} \to 0, & \frac{1}{2} < \beta < k. \end{cases}$$

For $\beta < \frac{1}{2}$, this is equivalent to $(\tilde{\lambda}^\star) \gg n^{-2(k-\beta)/(1-2\beta)}$; for $\beta = \frac{1}{2}$, any polynomial decay of $\tilde{\lambda}^\star$ ensures $n^{-2}\log(1/\tilde{\lambda}^\star) \to 0$; for $\beta > \frac{1}{2}$, (D) reduces to $n^{-2} \to 0$ and imposes no additional restriction.

We minimize $MSE(\tilde{\lambda}^\star) := \text{Det}(\tilde{\lambda}^\star) + \text{Stoch}(\tilde{\lambda}^\star)$ by differentiation.

**Regimes (by deterministic split $a + 2\beta - k$ and stochastic split $\beta = \frac{1}{2}$, with $\beta < k$).** Let

$$p := \frac{a+k-1}{k-\beta}.$$

(i) $a + 2\beta - k < 1$ (**Det** $\asymp (\tilde{\lambda}^\star)^p$)

- $\beta < \frac{1}{2}$ (Stoch $\asymp n^{-1}(\tilde{\lambda}^\star)^{-\frac{1-2\beta}{k-\beta}}$).

$$\text{MSE}(\tilde{\lambda}^\star) = (\tilde{\lambda}^\star)^p + n^{-1}(\tilde{\lambda}^\star)^{-\frac{1-2\beta}{k-\beta}}, \quad p(\tilde{\lambda}^\star)^{p+\frac{1-2\beta}{k-\beta}} = n^{-1}.$$

  Hence
$$\tilde{\lambda}^\star \asymp n^{-\frac{k-\beta}{a+k-2\beta}}, \qquad \text{MSE} \asymp n^{-\frac{a+k-1}{a+k-2\beta}}.$$

- $\beta = \frac{1}{2}$ (Stoch $\asymp n^{-1}\log(1/\tilde{\lambda}^\star)$).

$$\text{MSE}(\tilde{\lambda}^\star) = (\tilde{\lambda}^\star)^p + n^{-1}\log(1/\tilde{\lambda}^\star), \quad p(\tilde{\lambda}^\star)^p \asymp n^{-1}.$$

  Thus
$$\tilde{\lambda}^\star \asymp n^{-\frac{k-\beta}{a+k-1}}, \qquad \text{MSE} \asymp n^{-1}\log n.$$

- $\frac{1}{2} < \beta < k$ (Stoch $\asymp n^{-1}$). Choose $\tilde{\lambda}^\star \to 0$ with $(\tilde{\lambda}^\star)^p = o(n^{-1})$; e.g.

$$\tilde{\lambda}^\star \ll n^{-\frac{k-\beta}{a+k-1}} \quad \Rightarrow \quad \text{MSE} \asymp n^{-1}.$$

(ii) $a + 2\beta - k = 1$ (**Det** $\asymp (\tilde{\lambda}^\star)^2 \log(1/\tilde{\lambda}^\star)$)

- $\beta < \frac{1}{2}$ (Stoch $\asymp n^{-1}(\tilde{\lambda}^{\star})^{-\frac{1-2\beta}{k-\beta}}$).

$$\text{MSE}(\tilde{\lambda}^{\star}) = (\tilde{\lambda}^{\star})^2 \log(1/\tilde{\lambda}^{\star}) + n^{-1}(\tilde{\lambda}^{\star})^{-\frac{1-2\beta}{k-\beta}}.$$

It yields

$$(\tilde{\lambda}^{\star})^{2+\frac{1-2\beta}{k-\beta}}\left(2\log(1/\tilde{\lambda}^{\star}) - 1\right) \asymp n^{-1}.$$

Hence

$$\tilde{\lambda}^{\star} \asymp n^{-\frac{k-\beta}{2k+1-4\beta}} (\log n)^{\frac{k-\beta}{2k+1-4\beta}}, \qquad \text{MSE} \asymp n^{-\frac{2(k-\beta)}{2k+1-4\beta}} (\log n)^{\frac{2(k-\beta)}{2k+1-4\beta}}.$$

- $\beta = \frac{1}{2}$ (Stoch $\asymp n^{-1}\log(1/\tilde{\lambda}^{\star})$). Solving $2\tilde{\lambda}^{\star} - \frac{1}{n\tilde{\lambda}^{\star}} = 0$ gives

$$\tilde{\lambda}^{\star} \asymp n^{-1/2}, \qquad \text{MSE} \asymp n^{-1}\log n.$$

- $\frac{1}{2} < \beta < k$ (Stoch $\asymp n^{-1}$). Choose $\tilde{\lambda}^{\star} \to 0$ with $(\tilde{\lambda}^{\star})^2 \log(1/\tilde{\lambda}^{\star}) = o(n^{-1})$; e.g.

$$\tilde{\lambda}^{\star} \ll n^{-1/2} \quad \Rightarrow \quad \text{MSE} \asymp n^{-1}.$$

**(iii)** $a + 2\beta - k > 1$ (**Det** $\asymp (\tilde{\lambda}^{\star})^2$)

- $\beta < \frac{1}{2}$ (Stoch $\asymp n^{-1}(\tilde{\lambda}^{\star})^{-\frac{1-2\beta}{k-\beta}}$). Balancing $(\tilde{\lambda}^{\star})^2$ and $n^{-1}(\tilde{\lambda}^{\star})^{-\frac{1-2\beta}{k-\beta}}$ gives

$$\tilde{\lambda}^{\star} \asymp n^{-\frac{k-\beta}{2k+1-4\beta}}, \qquad \text{MSE} \asymp n^{-\frac{2(k-\beta)}{2k+1-4\beta}}.$$

- $\beta = \frac{1}{2}$ (Stoch $\asymp n^{-1}\log(1/\tilde{\lambda}^{\star})$).

$$\tilde{\lambda}^{\star} \asymp n^{-1/2}, \qquad \text{MSE} \asymp n^{-1}\log n.$$

- $\frac{1}{2} < \beta < k$ (Stoch $\asymp n^{-1}$). Choose $\tilde{\lambda}^{\star} \to 0$ with $(\tilde{\lambda}^{\star})^2 = o(n^{-1})$; e.g.

$$\tilde{\lambda}^{\star} \ll n^{-1/2} \quad \Rightarrow \quad \text{MSE} \asymp n^{-1}.$$

**Result.** The optimal tuning $\tilde{\lambda}^{\star}$ and the resulting MSE are:

**(i)** $k - 2\beta > a - 1$ (equivalently $a + 2\beta - k < 1$):

$$\tilde{\lambda}^{\star} \asymp \begin{cases} n^{-\dfrac{k-\beta}{a+k-2\beta}}, & \beta < \dfrac{1}{2}, \\[2mm] n^{-\dfrac{k-\beta}{a+k-1}}, & \beta = \dfrac{1}{2}, \\[2mm] \ll n^{-\dfrac{k-\beta}{a+k-1}}, & \dfrac{1}{2} < \beta < k, \end{cases}$$

$$\text{MSE} \asymp \begin{cases} n^{-\dfrac{a-1+k}{(a-1)+(k+1)-2\beta}}, & \beta < \dfrac{1}{2}, \\[2mm] n^{-1}\log n, & \beta = \dfrac{1}{2}, \\[2mm] n^{-1}, & \dfrac{1}{2} < \beta < k. \end{cases}$$

**(ii)** $k - 2\beta = a - 1$ (equivalently $a + 2\beta - k = 1$):

$$\tilde{\lambda}^{\star} \asymp \begin{cases} n^{-\dfrac{k-\beta}{2k+1-4\beta}} (\log n)^{\dfrac{k-\beta}{2k+1-4\beta}}, & \beta < \dfrac{1}{2}, \\[2mm] n^{-1/2}, & \beta = \dfrac{1}{2}, \\[2mm] \ll n^{-1/2}, & \dfrac{1}{2} < \beta < k, \end{cases}$$

$$
\text{MSE} \asymp \begin{cases}
n^{-\dfrac{2(k-\beta)}{2k+1-4\beta}} (\log n)^{\dfrac{2(k-\beta)}{2k+1-4\beta}}, & \beta < \dfrac{1}{2}, \\[2ex]
n^{-1} \log n, & \beta = \dfrac{1}{2}, \\[2ex]
n^{-1}, & \dfrac{1}{2} < \beta < k.
\end{cases}
$$

**(iii)** $k - 2\beta < a - 1$ (equivalently $a + 2\beta - k > 1$):

$$
\tilde{\lambda}^\star \asymp \begin{cases}
n^{-\dfrac{k-\beta}{2k+1-4\beta}}, & \beta < \dfrac{1}{2}, \\[2ex]
n^{-1/2}, & \beta = \dfrac{1}{2}, \\[2ex]
\ll n^{-1/2}, & \dfrac{1}{2} < \beta < k,
\end{cases}
\qquad
\text{MSE} \asymp \begin{cases}
n^{-\dfrac{2(k-\beta)}{2k+1-4\beta}}, & \beta < \dfrac{1}{2}, \\[2ex]
n^{-1} \log n, & \beta = \dfrac{1}{2}, \\[2ex]
n^{-1}, & \dfrac{1}{2} < \beta < k.
\end{cases}
$$

To consider all the function in the RKHS, we can set $a \to 1$ to find the rate of all the functions in the RHKS. Assume $k > 1$ and $\beta < k$. Then the asymptotic orders of the optimal tuning $\tilde{\lambda}^*$ and the resulting mean squared error are:

$$
\begin{cases}
\beta < \frac{1}{2}: & \tilde{\lambda}^* \asymp n^{-\frac{k-\beta}{k+1-2\beta}}, \qquad \text{MSE} \asymp n^{-\frac{k}{k+1-2\beta}}, \\[2ex]
\beta = \frac{1}{2}: & \tilde{\lambda}^* \asymp n^{-\frac{k-\frac{1}{2}}{k}}, \qquad \text{MSE} \asymp n^{-1} \log n, \\[2ex]
\beta > \frac{1}{2}: & \tilde{\lambda}^* \lesssim n^{-1/2}, \qquad \text{MSE} \asymp n^{-1}.
\end{cases}
$$

## C  PROOF OF LEMMA 1

*Proof.* Recall the definition of $\mathbf{K}_c$, an $L \times L$ kernel matrix and the kernel vector evaluated at the cluster centers are

$$
\mathbf{K}_c = \begin{pmatrix}
K(C_1, C_1) & \cdots & K(C_1, C_L) \\
\vdots & \ddots & \vdots \\
K(C_L, C_1) & \cdots & K(C_L, C_L)
\end{pmatrix}, \qquad
\mathbf{K}(x_0, C) = \begin{bmatrix}
K(x_0, C_1) \\
\vdots \\
K(x_0, C_L)
\end{bmatrix}.
$$

Let $\psi_1, \ldots, \psi_L$ denote the first $L$ eigenfunctions of the kernel function $K(\cdot, \cdot)$ evaluated on the centers and define

$$
\boldsymbol{\eta}_\ell := \begin{bmatrix}
\psi_\ell(C_1) \\
\vdots \\
\psi_\ell(C_L)
\end{bmatrix}, \qquad \ell = 1, \ldots, L.
$$

There exists an orthogonal matrix $\boldsymbol{Q}\,\boldsymbol{Q} = (\boldsymbol{\eta}_1, \cdots, \boldsymbol{\eta}_L)$, where $\boldsymbol{Q}^T \boldsymbol{Q} = \boldsymbol{Q}\boldsymbol{Q}^T = \boldsymbol{I}_L$, such that

$$
\boldsymbol{Q}^\top \mathbf{K}_c \boldsymbol{Q} = \begin{pmatrix}
\lambda_1 & & \\
& \ddots & \\
& & \lambda_L
\end{pmatrix} =: \Lambda.
$$

The vector $\mathbf{K}_{x_0}$ satisfies

$$
\boldsymbol{K}_{x_0} = \left(\boldsymbol{I} + L^{-1}\tilde{\lambda}^{\star-1}\boldsymbol{K}_c\right)^{-1} \boldsymbol{K}(x_0, C) = \begin{bmatrix}
K_{x_0 1} \\
\vdots \\
K_{x_0 L}
\end{bmatrix}.
$$

Using the eigen decomposition of $\mathbf{K}_c$,

$$
\left(\mathbf{I} + L^{-1}\tilde{\lambda}^{\star-1}\mathbf{K}_c\right)^{-1} = \boldsymbol{Q}\left(\mathbf{I} + L^{-1}\tilde{\lambda}^{\star-1}\Lambda\right)^{-1}\boldsymbol{Q}^T,
$$

the transformed kernel vector at $x_0$ is

$$\mathbf{K}_{x_0} = \boldsymbol{Q}\left(\mathbf{I} + L^{-1}\tilde{\lambda}^{\star-1}\Lambda\right)^{-1}\boldsymbol{Q}^\top\mathbf{K}(x_0, C).$$

Since

$$\boldsymbol{Q}^\top\mathbf{K}(x_0, C) = \begin{bmatrix} \boldsymbol{\eta}_1^\top\mathbf{K}(x_0, C) \\ \vdots \\ \boldsymbol{\eta}_L^\top\mathbf{K}(x_0, C) \end{bmatrix},$$

the $l$-th coordinate of $\mathbf{K}_{x_0} = \left(K_{x_0 1}, \cdots, K_{x_0 L}\right)^\top$ is

$$K_{x_0 l} = \sum_{r=1}^{L} \frac{\boldsymbol{\eta}_r^\top\mathbf{K}(x_0, C)}{1 + L^{-1}\tilde{\lambda}^{\star-1}\lambda_r}\,\psi_r(C_l), \qquad l = 1, \dots, L. \tag{9}$$

The cluster-level resampling weight is defined as

$$\omega_{x_0, l, C} = \frac{|K_{x_0 l}|}{\sum_{m=1}^{L} |K_{x_0 m}|},$$

and each sample $x_i \in \mathcal{C}_l$ receives

$$\omega_i(x_0) = N_l^{-1}\,\omega_{x_0, l, C}.$$

Then we can write

$$\omega_{x_0, l, C} = \frac{\displaystyle\sum_{r=1}^{L} \frac{L^{-1}\boldsymbol{\eta}_r^\top\mathbf{K}(x_0, C)}{1 + L^{-1}\tilde{\lambda}^{\star-1}\lambda_r}\,\psi_r(C_l)}{\displaystyle\sum_{m=1}^{L}\sum_{r=1}^{L} \frac{L^{-1}\boldsymbol{\eta}_r^\top\mathbf{K}(x_0, C)}{1 + L^{-1}\tilde{\lambda}^{\star-1}\lambda_r}\,\psi_m(C_m)}, \qquad l = 1, \dots, L.$$

Note that

$$L^{-1}\boldsymbol{\eta}_r^\top\mathbf{K}(x_0, C) = L^{-1}\sum_{q=1}^{L}\psi_r(C_q)\,K(x_0, C_q) \approx \int \psi_r(t)\,K(x_0, t)\,dt = \lambda_r\psi_r(x_0),$$

and substituting this approximation into the numerator of $\omega_{x_0, l, C}$ gives

$$\sum_{r=1}^{L} \frac{L^{-1}\boldsymbol{\eta}_r^\top\mathbf{K}(x_0, C)}{1 + L^{-1}\tilde{\lambda}^{\star-1}\lambda_r}\,\psi_r(C_l) \approx \frac{1}{L}\sum_{r=1}^{L} \frac{\lambda_r}{1 + L^{-1}\tilde{\lambda}^{\star-1}\lambda_r}\,\psi_r(x_0)\psi_r(C_l).$$

Similarly, the denominator becomes

$$\sum_{m=1}^{L}\sum_{r=1}^{L} \frac{L^{-1}\boldsymbol{\eta}_r^\top\mathbf{K}(x_0, C)}{1 + L^{-1}\tilde{\lambda}^{\star-1}\lambda_r}\,\psi_r(C_m) \approx \frac{1}{L}\sum_{m=1}^{L}\sum_{r=1}^{L} \frac{\lambda_r}{1 + L^{-1}\tilde{\lambda}^{\star-1}\lambda_r}\,\psi_r(x_0)\psi_r(C_m).$$

So

$$\omega_{x_0, l, C} = \frac{\displaystyle\sum_{r=1}^{L} \frac{\lambda_r}{L\tilde{\lambda}^{\star} + \lambda_r}\,\psi_r(x_0)\psi_r(C_l)}{\displaystyle\sum_{m=1}^{L}\sum_{r=1}^{L} \frac{\lambda_r}{L\tilde{\lambda}^{\star} + \lambda_r}\,\psi_r(x_0)\psi_r(C_m)}, \qquad l = 1, \dots, L.$$

Define a new kernel $\widetilde{K}$ by the following

$$\widetilde{K}(x, y) = \sum_{r=1}^{\infty} \frac{\lambda_r}{L\tilde{\lambda}^{\star} + \lambda_r}\,\psi_r(x)\psi_r(y),$$

so that

$$\omega_{x_0,l,C} \asymp \frac{\widetilde{K}(x_0, C_l)}{\sum_{m=1}^{L} \widetilde{K}(x_0, C_m)}.$$

Each sample $x_i \in \mathcal{C}_l$ receives weight

$$\omega_i(x_0) = N_l^{-1} \omega_{x_0,l,C},$$

so

$$\omega_{x_0,j} = \mathbb{E}\left\{ \sum_{i=1}^{N} \omega_i(x_0)\, \psi_j^2(X_i) \right\} = \mathbb{E}\left\{ \sum_{l=1}^{L} \sum_{i \in \mathcal{C}_l} N_l^{-1}\, \omega_{x_0,l,C}\, \psi_j^2(X_i) \right\}.$$

Because $\omega_{x_0,l,C}$ is constant within $\mathcal{C}_l$, we have

$$\omega_{x_0,j} = \mathbb{E}\left[ \sum_{l=1}^{L} \omega_{x_0,l,C}\, \mathbb{E}\left\{ N_l^{-1} \sum_{i \in \mathcal{C}_l} \psi_j^2(X_i) | \mathcal{C} \right\} \right].$$

The within–cluster average may be approximated by the value at the center:

$$\mathbb{E}\left\{ N_l^{-1} \sum_{i \in \mathcal{C}_l} \psi_j^2(X_i) \right\} \approx \psi_j^2(C_l),$$

hence

$$\omega_{x_0,j} \asymp \mathbb{E}\left\{ \sum_{l=1}^{L} \omega_{x_0,l,C}\, \psi_j^2(C_l) \right\}. \tag{10}$$

As $L \to \infty$ and the centers $\{C_l\}$ form a dense design from the marginal $\pi$, the sums in (equation 10) converge to integrals:

$$\omega_{x_0,j} \asymp \frac{\int \widetilde{K}(x_0, c)\, \psi_j^2(c)\, d\pi_c}{\int \widetilde{K}(x_0, c)\, d\pi_c}.$$

Applying the Cauchy–Schwarz inequality (e.g. Theorem 1 in Shi et al. (2009)), for kernels $K(x, y)$ such that $\int K^2(x, y) d\pi(y) < \infty$, the $j$-th eigenfunction of $K(x, y)$ is bounded by $|\psi_j(x)| \leq C/\lambda_j$ for some constant $C$. If $\lambda_j \asymp j^{-k}$, then it implies that $\omega_{x_0,j} \leq \lambda_j^{-2} \asymp j^{2k}$. $\qquad\square$

# D   ADDITIONAL NUMERICAL RESULT

## D.1   CLUSTERING METHOD ANALYSIS

To investigate the effect of clustering methods on the proposed estimator, we conduct additional experiments by testing different clustering while holding all other components fixed. Using the same dataset and retaining all 90 acoustic features under a Gaussian RBF kernel, we evaluate three clustering methods strategies for each $(N, n)$ configuration: K-means (Lloyd's algorithm with k-means++ initialization), random projection K-means, and K-medoids, a PAM-style medoid method that is more robust to outliers and non-spherical clusters (Kaufman & Rousseeuw, 1990).

Table 6: Comparison the effect of clustering methods including K-means, random projection K-means (RP-kmeans) and K-mediods, on the proposed method using all the 90 features of the YearPredictionMSD dataset. Each entry reports (RMSE / MSE). Best MSE within each $(N, n)$ configuration is highlighted in bold.

| | | K-means | | RP-kmeans | | K-medoids | |
|---|---|---|---|---|---|---|---|
| $N$ | $n$ | RMSE | MSE | RMSE | MSE | RMSE | MSE |
| 2000 | 500 | 9.789 | 95.83 | 9.856 | 97.15 | **9.734** | **94.75** |
| 5000 | 1000 | 9.734 | 94.75 | **9.739** | **94.75** | 9.761 | 95.29 |
| 5000 | 2000 | 9.812 | 96.20 | 9.811 | 96.26 | **9.728** | **94.51** |
| 10000 | 1000 | **9.246** | **85.45** | 9.356 | 87.51 | 9.325 | 86.96 |
| 10000 | 2000 | 9.149 | 83.70 | 9.090 | 82.62 | **9.073** | **82.31** |
| 20000 | 1000 | 9.374 | 87.89 | 9.409 | 88.89 | **9.289** | **86.28** |
| 20000 | 2000 | **8.872** | **80.48** | 8.976 | 80.55 | 9.024 | 81.43 |
| 50000 | 1000 | 9.360 | 87.61 | 9.499 | 90.24 | **9.339** | **87.20** |
| 50000 | 2000 | 9.196 | 84.56 | **9.117** | **83.12** | 9.126 | 83.28 |

## E    NUMERICAL EXPERIMENT SETUP

### E.1    ADDITIONAL DETAILS FOR SECTION 4.2

**Data generation.**    Let $X = (X_1, \ldots, X_{20})^\top$ with $X_j \overset{\text{iid}}{\sim} \text{Unif}[0, 1]$. The response is

$$Y = f_0(X_1, \ldots, X_5) + \epsilon, \qquad f_0(x) = 10 \sin(\pi x_1 x_2) + 20(x_3 - 0.5)^2 + 10x_4 + 5x_5,$$

with $\epsilon \sim \mathcal{N}(0, 0.1^2)$. Only the first five coordinates influence the signal. For each replicate we draw a training set of size $N \in \{2000, 5000, 10000\}$ and evaluate on a fixed test set of size 100. We run 100 replications for each $(N, n)$, with $n \in \{100, 500, 1000\}$.

**Hyperparameter tuning.**    All methods use the RBF kernel and an $8\%$ hold-out for tuning.

**FALKON (2017).**    Nyström landmarks are chosen uniformly with $M = n$. Bandwidth is selected from $\{\gamma_0/2, \gamma_0, 2\gamma_0\}$, where $\gamma_0$ is the median heuristic from the hold-out set. The ridge parameter is chosen from $\{0.5, 1.0, \ldots, 4.0\}/N$. We run 300 PCG iterations.

**FALKON (2020).**    The PyTorch/KeOps implementation of Meanti et al. (2020) uses the same landmark scheme, bandwidth grid, ridge grid, and $8\%$ hold-out. Only MSE is reported due to cross-platform runtime differences.

**Proposed method.**    We resample $n$ points using the proposed weights, and average predictions over $B = 3$ subsets. Each replicate applies MATLAB k-means (k-means++ initialization) to compute $n$ cluster centers. The center bandwidth is the median squared inter-center distance; the subset bandwidth is set to twice this value. We tune $\tilde{\lambda}^*$ over $\{10^{-4}, 10^{-3}, \ldots, 10^1\}$.

**Nyström KRR.**    We use $M = n$ random landmarks and tune the ridge parameter over $\{10^{-4}, 10^{-3}, \ldots, 10^1\}$ using the same hold-out split.

### E.2    ADDITIONAL DETAILS FOR THE REAL-DATA STUDY

**Data and preprocessing.**    We use the standard split of the YEARPREDICTIONMSD dataset: the first 463,715 rows form the training pool and the remaining observations form the test pool. For each experiment we sample $N \in \{2000, 5000, 10000, 20000\}$ from the training pool and use a fixed test set of size 1000. All randomization is seeded (rng(42)).

**Feature selection.** We rank the 90 features by absolute Pearson correlation with the response and keep the top $p \in \{30, 60, 90\}$, where $p=90$ means using all features.

**Proposed method.** We compute 500 clustering centers using RP-$k$-means (random projection to 32 dimensions followed by $k$-means). At estimation time we draw $B = 3$ independent subsets of size $n$, with bandwidths set by the median-distance rule. We use $\lambda_{\text{tuning}} = 1$ and report results for three kernels: Gaussian RBF (Prop(G)), Matérn-3/2 (Prop(M)), and Laplace (Prop(L)). Runtime includes center computation and the sum of the three kernel runs.

**FALKON.** We draw $M$ uniform Nyström landmarks and run the PCG solver for 20 iterations with regularization $\lambda = 10^{-6}$. The kernel is always Gaussian RBF with $\sigma = 6$ following Rudi et al. (2017).

**Nyström KRR.** We sample $M$ uniform landmarks, compute a low-rank decomposition of $K_{ZZ}$, and solve ridge regression using the same $(M, \gamma, \lambda)$ as in FALKON, again using only the Gaussian RBF kernel.

**Evaluation.** For all methods we report test MSE and total wall-clock time, including tuning, subsampling, clustering, feature preprocessing, and model training.

### E.3 TIME AND MEMORY COMPLEXITIES

We briefly sketch how the orders in Table 5 are obtained.

**FALKON (Rudi et al., 2017).** Standard Nyström KRR with $M$ landmarks has time cost $O(NM) + O(M^3)$ and memory cost $O(NM + M^2)$. The FALKON analysis in Rudi et al. (2017) chooses the theoretically optimal $M \asymp \sqrt{N}$, which yields

$$O(NM) + O(M^3) \;=\; O(N^{3/2}) + O(N^{3/2}) \;=\; O(N^{3/2}),$$

with memory $O(NM + M^2) = O(N)$.

**FALKON (Meanti et al., 2020).** The large-scale implementation of Meanti et al. (2020) treats $M$ as a tuning parameter and uses an iterative solver with per-iteration cost $O(NM) + O(M^2)$ and memory $O(NM + M^2)$. Since $M$ is not tied to $\sqrt{N}$ in theory, the generic $O(NM) + O(M^2)$ form is reported in Table 5.

**Proposed + FALKON.** Let $n$ be the subdata size selected by the proposed method. The clustering-based selection uses $K$ clusters and $t$ iterations of k-means. At each iteration the cost is $O(nK)$, so the total time is $O(nKt)$ and memory is $O(n+K)$ for storing the data assignments and centers. Applying a Nyström–type solver (such as FALKON) to the subdata has computational cost $O(nM) + O(M^2)$ in time and $O(nM + M^2)$ in memory. Theorem 2 shows that the proposed estimator achieves the optimal full-data rate in Theorem 1 when

$$n \;\asymp\; N^{\frac{k+1-2\beta}{k+1}} \quad (\beta \in [0, 1/2)), \qquad n \;\asymp\; N^{\frac{k}{k+1}} \quad (\beta > 1/2).$$

Writing $n \asymp N^\gamma$ for these two regimes, the combined selection + solver cost becomes

$$\text{Time: } O(N^\gamma M + N^\gamma K t), \qquad \text{Memory: } O(N^\gamma M + M^2 + K),$$

which matches the expressions reported for the Proposed + FALKON method in Table 5.

