# OpenReview forum: "Optimal Sub-data Selection for Nonparametric Function Estimation in Kernel Learning with Large-scale Data"
_ICLR.cc/2026/Conference — Submitted to ICLR 2026_

### Official Review · Reviewer_c2tu · 2025-10-21

**Soundness:** 2
**Presentation:** 1
**Contribution:** 3
**Rating:** 4
**Confidence:** 3

**Summary:**

This paper deals with the problem of estimating a nonparametric function from a given reproducing kernel Hilbert space, when using large-scale (possibly with large sample size) training data.
This is a recurring problem in the field.
Specifically, the work focuses on a kernel Ridge Regression (KRR) problem where the RKHS is assumed given. Then, the goal is to estimate the kernel expansion coefficients and regularization weight (called tuning parameter in the paper) in a data-scalable manner.
For that, the proposed method looks for a fixed number of clustered data subsets whose centers act as a representative values of all the data samples in the cluster, and then solves the KRR problem over the resulting (smaller) data.
Finally, a theoretical discussion about the rate of convergence of the proposed method and experimental results are presented.

**Strengths:**

The presented approach seems novel and is well motivated.

**Weaknesses:**

**W1**: To me, the biggest weakness of this paper is an inconsistent, and sometimes vague, mathematical notation.
This makes following the main claims of the paper a tedious task.
Overall, this paper would benefit substantially from a consistent and properly introduced mathematical notation.

To give some examples: \
**W1.1**: The equation in line 132 defines the optimal argument as $\hat{f}\_{N,\lambda\_T}$ .
In the ensuing sections the comma in the subindex disappears. \
**W1.2**: In the expression in line 140, the authors use brackets [ ] to denote a vector. Later, vectors are denoted with parenthesis ( ), e.g., line 177, or curly braces { }, e.g., line 195, instead. Same for the transpose symbol, sometimes is an apostrophe $'$, e.g., line 266, and sometimes a superscript $^T$, e.g., line 268. \
**W1.3**: Some notation is introduced without definition, such as $\hat{f}\_{s\lambda}\^\*$ in line 151, and $\theta_i$ and the subscript $\_{n*n}$ in line 226.

I believe there are also some minor typos/mistakes such as: \
**W1.4**: The sentence spanning lines 158 to 161 defines the centers of the clusters redundantly. \
**W1.5**: In line 192, $\bar{\epsilon}_i$ is using the definition intended for $\bar{y}_i$. \
**W1.6**: In line 196 the $(i,j)$-th element is a set. It should likely be $K(C_i,C_j)$ or a singleton. \
**W1.7**: In line 277 $\mathscr{S}(x_0)$ is defined as a set of $x^*_i$ values; however, in line 279 it is used as the set of indices of those values. \
**W1.8**: There is typo in line 476, ``can effectively use~~s~~''.

**W2**: A discussion of the memory and time complexity comparing the proposed method to alternatives (e.g., Nyström and FALKON) would have been appreciated.
For instance, in a comparison table.

**Questions:**

**Q1**: In table 3, I assume that ``centers-time'' refers to the execution time of the clustering algorithm. Taking that into account, the computation time of the proposed method is still one, or even two, orders of magnitude larger than FALKON and Nyström.
Can you elaborate on that?

---

> ### Author Response · Authors · 2025-11-20
>
> We greatly appreciate the valuable suggestions and constructive comments from the reviewer. We have carefully addressed each point, and all major changes in the revised manuscript are highlighted in blue.
>
> W1. Inconsistent and vague mathematical notation
>
> We sincerely appreciate the reviewer for careful reading and the detailed feedback.
> We carefully reviewed all points raised and revised the manuscript thoroughly to ensure
> consistency in notations and presentations. Below is a summary of the main changes.
>
> • The notation for $\hat{f}_{N,\lambda_T}$ is now used consistently in all sections.
>
> • Vector notation has been standardized to use bracke $[\cdot]$ throughout.
>
> • The transpose operator is unified as the superscript ${}^T$ in all occurrences.
>
> • Undefined symbols such as $\hat{s}k^A$, $\theta_i$, and $n{nn}$ have been corrected.
>
> • Redundant definitions of cluster centers have been consolidated into a single clear description.
>
> • The previous typo involving $\bar{\xi}_i$ (now $\bar{\epsilon}_i$ in the revised draft) has been corrected.
>
> • The notation for the $(i,j)$-th entry of the center kernel matrix has been fixed to $K(C_i, C_j)$.
>
> • The definition and use of $\mathscr{S}(x_0)$ have been aligned so that it always denotes the set of selected indices.
>
> • The grammatical typo “can effectively uses’’ has been corrected.
>
> We also performed an additional pass over the entire manuscript to remove several other minor inconsistencies in notation and to improve overall clarity. We believe these changes substantially improve the readability and precision of the paper. We apologize for the earlier inconsistencies and any confusion they may have caused.
>
> W2. Discussion of time and memory complexity
>
> We appreciate your constructive suggestions. In this revision, we include a discussion of the memory and computation complexity (time) of our method when FALKON is applied to sub-data selected by our method. We added a new Table 5 at the end of Section 5.2 to discuss the time and memory complexity of the proposed method.
>
> Q1. Clarification of computation time in Table 3
>
> Thank you for the comments. This paper focuses on selecting a small subsample from a large scale data.To demonstrate the idea, we performed pointwise prediction, where for each test point we selected a relatively small dataset. Because of this reason, the computation time reported in Table 3 was for 1000 test data points, which is more favorable to sketching methods (such as FALKON and Nyström) that do the prediction using all test data points together.
>
> To demonstrate the advantage of the proposed method, in this revision, we combined the proposed method with FALKON to perform pointwise estimation by first selecting a subdata from the full data and apply FALKON to make pointwise prediction. By doing so, we can see that our proposed method can further improve the computation time of FALKON while maintaining the same statistical efficiency. See Table 4 of the revised paper.

---

> > ### Comment · Reviewer_c2tu · 2025-11-24
> >
> > I acknowledge all corrections and clarifications made by the author(s).
> >
> > Regarding presentation, the paper has noticeably improved; however, I still respectfully think that some additional work is necessary.
> >
> > For instance, in line 269, the expression $\\mathscr{S}(x_0) = \\{i^\*_1, \\dots , i^\*_n \\} \\in \\{ 1, \\cdots, N \\}$ should use the subset symbol ( $\\subset$ or $\\subseteq$ ), and preferably use the same $\\dots$ symbol.
> > As currently written, the $\in$ symbol incorrectly implies that the entire set $\\{i^\*_1, \\dots , i^\*_n \\}$ is treated as a single element within the set of indices $\\{1,\dots, N\\}$.
> > While this typo alone may be irrelevant, the fact that it persists despite being addressed (as a response to my previous comment) signals that some careful additional proofreading/rewriting may be necessary.
> >
> > For this reason, I will maintain my current score.

---

> > > ### Author Response · Authors · 2025-11-26
> > >
> > > Thank you for your careful reading and constructive comments. We apologize for overlooking some notation issues despite our earlier revisions. To further improve consistency, we have conducted an additional round of thorough proofreading and revision. The major changes are as follows:
> > >
> > > 1. We changed all the lower case $x_i$ to $X_i$ to maintain consistent notation for the data.
> > > 2. We made all the expectation and variance notations consistent.
> > > 3. We standardized all occurrences of ellipsis notations. We now consistently use '$\ldots$' for lists such as $\{C_1, \ldots, C_L\}$, and use centered dots '$\cdots$' in vector or matrix contexts such as $[y_1,\cdots,y_n]^\top$.
> > > 4. We carefully corrected all bracket usage throughout the manuscript following the hierarchy: parentheses, curly braces, then square brackets.
> > >
> > > These corrections have been applied throughout the manuscript, and we have also addressed several additional grammatical and notation issues. We hope that these revisions have improved the overall clarity and consistency of the paper. Please check our revised manuscript for the above mentioned changes.

---

### Official Review · Reviewer_dGCQ · 2025-10-23

**Soundness:** 1
**Presentation:** 1
**Contribution:** 1
**Rating:** 2
**Confidence:** 4

**Summary:**

The paper introduces a subset selection method for large-scale nonparametric regression in reproducing kernel Hilbert spaces (RKHS). The main goal is to improve prediction accuracy when computational resources limit the use of all available data. The proposed approach first partitions the dataset using clustering methods such as k-means to identify representative centers, then assigns optimal sampling weights to clusters in order to minimize the pointwise mean squared error of the resulting kernel estimator. Weighted subsampling is performed multiple times, kernel ridge regression models are trained on each subset, and their predictions are averaged. The authors claim to achieve better MSE error compared with basic Nyström or FALKON (but its first version in 2017 in Matlab).

**Strengths:**

The topic is interesting and the paper targets the well-known bottleneck of kernel methods, i.e. scalability. The clustering and sampling procedure with optimal weights can be interesting, but a clearer and more precise exposition, and deeper experiments are needed to evaluate it.

**Weaknesses:**

### **Critical Observations**

1. **Outdated comparison**
   The paper compares only against the *original* **FALKON (2017)** and a basic **Nyström** baseline, but not against more recent or optimized implementations — for example, *“Kernel methods through the roof: handling billions of points efficiently”* (Meanti et al., NeurIPS 2020), which provides a **modern, fast version of FALKON**. As a result, the experimental comparison does not reflect the current state of the art.

2. **Misleading comparison and trivial findings**
   The comparison is conceptually weak. Works such as *“Less is more: Nyström computational regularization”* (Rudi et al., 2015) and *“FALKON: An optimal large scale kernel method”* (Rudi et al., 2017) have already shown that Nyström-based methods can **achieve optimal learning rates** while maintaining **computational efficiency**, provided that a **sufficient number of landmarks** are used (e.g., √n in kernel ridge regression).
   It is therefore **expected** that using too few subsampled points degrades accuracy, and that more sophisticated (but slower) sampling strategies can improve MSE. Even a fully greedy point-selection scheme — which maximizes accuracy at each step — would outperform uniform sampling, but at a much higher computational cost.
   -> Consequently, the reported “superior accuracy” is **trivial and somewhat misleading**: it arises from sacrificing scalability, missing the main purpose of FALKON, which is to remain **fast while preserving accuracy** once enough centers are used.

3. **Computational inefficiency**
   The proposed method is **consistently slower** than FALKON (often by an order of magnitude) while achieving only **moderate MSE improvements**.
   -> The key insight of FALKON — maintaining accuracy while being computationally efficient — is not addressed or appreciated in this work.

4. **Clarity and presentation issues**
   The paper is **poorly structured and very difficult to follow**. Among all, definitions and notations are often unclear or inconsistent:
   - The meaning of \( y_{ij} \) (double index) is never explained.
   - The notation \( \hat{w}_{x,i,C} \) contains an unexplained hat (possibly a typo).
   - \( K_{xi} \) is defined *after* Equation (2) but used *before* it.
More than that, theoretical assumptions are **scattered and vaguely presented**, often embedded directly within theorem statements without being clearly introduced and discussed.  How do these assumptions compare with the rest of the literature (which is **largely** missing in the entire paper)?

   Overall, the exposition lacks precision and logical flow, making the paper unnecessarily hard to read and evaluate.

**Questions:**

Most perplexities are already expressed above under Weaknesses.
- Why is the most recent literature (both theory and algorithms) not considered?
- in Results at pag. 8 it's said that the method "remains efficient", but it is 100x slower than old version of FALKON, how can be considered efficient when datasets contain billions of points?

---

> ### Author Response · Authors · 2025-11-20
>
> We greatly appreciate the valuable suggestions and constructive comments from the reviewer and have carefully addressed each point. All major changes in the revised manuscript are highlighted in blue.
>
> W1. Thanks for bringing the reference to our attention. We have now included additional experiments in both simulation and real data analysis using the FALKON code from “Kernel Methods Through the Roof: Handling Billions of Points Efficiently” (Meanti et al., 2020). These new experiments follow the same data-generation process, tuning strategy, and overall experimental design as in our original study. We have added the results using this modern FALKON implementation in both Tables 2 and 3 of the revised manuscript.
>
> W2. We agree that FALKON and Nyström could achieve the best learning rate if sufficient landmarks are used, while speeding up the computation time. The superior computation speed of FALKON and Nyström was shown in our simulation and real data studies. Because the number of landmarks ($n=M$ in our notation) chosen for the simulation and real data analysis are more than $\sqrt{N}$ ($N$ is the entire sample size), we believe that the results presented in Tables 2 and 3 are appropriate and not misleading. However, to address your concerns, we included the FALKON in Meanti et al. (2020) in our updated real data study (see the updated Table 3). We observe it performed better than other methods when $N=20000$ and $n=2000$.
>
> It is worthwhile to note that the objective of our method is different from sketching methods (such as FALKON and Nyström) which are designed to sample columns of a kernel matrix. Our method is designed to select an informative subdata from the entire data so that it can maintain the same asymptotic efficiency. Theorem 2 in the paper suggests that an informative subdata of size $N^{(k+1-2\beta)/(k+1)}$ if $\beta \in [0,1/2)$ (or $N^{k/(k+1)}$ if $\beta>1/2$) would be sufficient to maintain the statistical efficiency.
>
> Regarding the scalability issue, our method can be combined with Nyström-based methods (such as FALKON) to speed up computation. This can be achieved by first obtaining an informative sub-data using our method and then applying FALKON to the selected sub-data. This combination could improve the computation speed while maintaining the same asymptotical statistical efficiency. This integrated method can reduce time to the order of $O(N^{\gamma} M)$, $\gamma<1$ and $M$ is the number of landmarks, which is smaller than $O(NM)$ when using FALKON alone. To demonstrate this, we further implemented a combination of our method and FALKON to demonstrate the improvement in computation time while maintaining similar statistical efficiency. See updated Table 4 for the new results.
>
> W3. Thank you for your comments. This paper focuses on selecting a small subsample from a large scale data. In Table 4, we performed pointwise prediction for each test point, where a relatively small subdata is selected for each test point. Because of this reason, the computation time reported in Table 3 was for 1000 test data points, which is more favorable to sketching methods (such as FALKON and Nyström) that do bulk prediction using one entire data set.
>
> To demonstrate the effectiveness of our method in improving computation time for pointwise prediction, we have combined the proposed method with some existing sketching algorithms such as FALKON to reduce the computational time. It is shown in Table 4 that the proposed method is able to further reduce the computation time of FALKON while maintaining the same statistical efficiency. A numerical implementation of this method is presented in Table 4, which has demonstrated the complementary between our method and FALKON in improving computation time.
>
> W4. We have carefully checked and updated our definitions and notations to make them clear and consistent. The assumptions on eigenvalues and eigenfunctions in Theorems 1 and 2 are standard in the literature of RKHS estimators (e.g., "A reproducing kernel Hilbert space approach to functional linear regression" (Yuan & Cai, 2010) and "Divide and conquer kernel ridge regression: a distributed algorithm with minimax optimal rates" (Zhang et al., 2015)). To make this point clear, we have included some comments after Theorem 1. Moreover, we added a Lemma 1 to justify the assumption on the weight function $w_{x_0,j}$ in Theorem 2.
>
> Q1. Our method focuses on sub-data selection, we have included most recent literature for sub-data selection methods and we have tried to include most recent sketching methods, but we may miss some sketch methods. Thank you for bringing them to our attention. We have included FALKON (Meanti et al., 2020) and compared with it in both simulation studies and real data study.
>
> Q2. In this revision, we implemented a combination of our sub-data selection method with FALKON. The combination can further improve the computation speed of FALKON. See our responses to your comment W3.

---

### Official Review · Reviewer_Xqhh · 2025-10-31

**Soundness:** 3
**Presentation:** 1
**Contribution:** 1
**Rating:** 2
**Confidence:** 4

**Summary:**

This paper addresses the computational challenge of large-scale nonparametric function estimation in Reproducing Kernel Hilbert Spaces (RKHS) by proposing a weighted sub-data selection method. Theoretical results show that the proposed method achieves a faster convergence rate than simple random sampling (SRS).

**Strengths:**

S1. The paper provides a theoretical analysis, demonstrating that its proposed estimator achieves a convergence rate superior to Simple Random Sampling.

S2. The method demonstrates superior performance in MSE compared to established benchmarks.

**Weaknesses:**

W1. The technical components of the proposed method rely on established algorithms and principles that lack innovation. For instance, the core clustering step relies on k-means, a classical algorithm initially proposed more than six decades ago (e.g., by Stuart Lloyd in 1957). Similarly, the use of variance minimization to derive optimal weights is a long-standing principle in classical optimization and statistics. While the integration of these elements for point-wise prediction in RKHS is applied to a specific setting, the overall methodology constitutes a recombination of existing techniques rather than a conceptually novel contribution. Furthermore, the problem being addressed, scaling kernel methods,does not align with pivotal frontier challenges in contemporary AI research, thereby limiting the broader impact and relevance of this work.

W2. The paper suffers from several issues in scholarly rigor and exposition that impact its professionalism and readability, for example:
1. In Section 2, the theoretical foundation lacks references to crucial theorems like Mercer's theorem and the Representer Theorem. Their absence makes the theoretical setup incomplete.
2. The meaning of notation is sometimes unclear. For instance, in Section 2.1, the expression {K(x, C_1), ...} does not specify whether the braces {} denote a set or a vector, creating unnecessary confusion.
3. Incorrect formatting of references is present throughout the manuscript, which diminishes the work's professionalism and adherence to standard academic conventions.

**Questions:**

Q1. In the first part of Section 2, the parameter $\lambda_T$ is introduced. Could you please clarify the relationship and distinction between $\lambda_T$ and the regularization parameter lambda used later in Section 2.1? Specifically, what does the subscript $T$ denote, and what is the conceptual or mathematical difference between these two symbols?

Q2.The proposed sampling strategy appears to rely on a clustering step. Could you please discuss the potential sensitivity of your method's final performance to the specific clustering algorithm chosen? Have you experimented with different clustering methods?

---

> ### Author Response · Authors · 2025-11-20
>
> We greatly appreciate the valuable suggestions and constructive comments from the reviewers.
> We have devoted significant time and effort to carefully address each of them and have made revisions based on your comments and questions. For clarity, all major changes in the revised manuscript are highlighted in blue.
>
>
> W1. Discussion of Novelty and Methodological Integration
>
> We thank the reviewer for the thoughtful comments.
>
> A growing body of recent work has demonstrated theoretical equivalences and connections between deep neural networks (DNNs) and kernel learning methods (e.g., “Neural tangent kernel: convergence and generalization in neural networks” (Jacot et al., 2018), “Transition to linearity of general neural networks with directed acyclic graph architecture” (Zhu et al., 2022), “Improving Node Classification with Neural Tangent Kernel: A Graph Neural Network Approach” (Zhang et al., 2024)).
> As highlighted in “To Understand Deep Learning We Need to Understand Kernel Learning” (Belkin et al., 2018), gaining a deeper understanding of more tractable kernel methods is an essential step toward developing a solid theoretical foundation for DNNs. From this perspective, we believe our research contributes to a topic of significant contemporary interest in AI, and one that aligns well with the themes of ICLR 2026. To address your comments, we included the above discussion to Section 6.
>
> While individual components of our approach, such as clustering techniques and the minimum-variance principle, already exist in the literature, one of our contributions lies in integrating these tools to address subdata selection in kernel learning. Moreover, how to use these techniques in kernel learning to choose informative subdata has not been investigated, and its impact on the statistical efficiency is unknown in the literature. Our theoretical analysis shows that the method leads to an improved convergence rate for estimation and prediction (see Theorem 2 in the paper).
> Our numerical studies further demonstrate that the proposed method achieves superior performance and it can be further combined with the state-of-the-art sketching algorithms to speed up kernel learning without sacrificing statistical efficiency (see Table 4 for the updated results for this combination).
>
>
> W2. Notation and Reference Formatting Issue
>
> Thank you for raising these issues and we apologize for the notation confusion.
> (i) We have added the missing references to key foundational theorems such as Mercer's
> theorem and the Representer Theorem.
> (ii) We carefully corrected notation inconsistencies throughout the manuscript; in
> particular, we now use $[\cdot]$ for vectors and {$\cdot$} for sets to avoid ambiguity.
> (iii) We thoroughly reviewed and corrected formatting issues in the bibliography to
> ensure consistency with standard academic conventions.
>
>
> Q1. Clarification of the Tuning Parameters $\lambda_T$ vs $\lambda$
>
> Thank you for your question. In the earlier draft, we used $\lambda$ to denote a general tuning parameter, where $T$ was used to denote ``tuning''. We are sorry about the confusion.
> To make our notations consistent, in the revised version, we have changed all $\lambda_T$ and $\lambda_T^\prime$ to $\lambda^*$. For the full data method, we use $\lambda^{\star}$ and use $\tilde{\lambda}^{\star}$
> to denote a tuning parameter for  the proposed method.
>
> Q2. Sensitivity to Clustering Algorithms
>
> We appreciate the reviewer’s constructive suggestion.
> We have tried different clustering methods in the revised paper including K-means, random projected K-means and k-medoids. See Section D.1 of the Appendix of the revised paper. We found that the proposed method performed similarly for different clustering methods. No single method dominates for all the scenarios. This demonstrates the robustness of the proposed method to clustering methods.

---

### Official Review · Reviewer_bZ3f · 2025-11-04

**Soundness:** 3
**Presentation:** 3
**Contribution:** 3
**Rating:** 6
**Confidence:** 4

**Summary:**

This paper studies kernel ridge regression (KRR) in reproducing kernel Hilbert spaces (RKHS) in the context of large sample sizes (N) and proposes a sub-data selection scheme aimed at matching or improving the statistical accuracy while simultanesouly reducing computational complexity. The method clusters inputs into L groups (via $k$-means or variants), chooses a tuning parameter using only the cluster centers, and then, for each test location $x$, assigns sampling probabilities to training points proportional to $ | K_x|$ (a transformed kernel vector that depends on $x$, the chosen kernel, the center kernel matrix, and the tuning parameter $\lambda$). The estimator averages KRR fits over \$B\$ resampled subsets of size $n$ much smaller than the sample size $N$. The paper provides a clean derivation of KRR rates and optimal tuning parameter value $\lambda_T\$ under eigenvalue decay (Theorem 1) and an asymptotic IMSE rate for the proposed estimator (Theorem 2) showing improvement over simple random sampling (SRS) under assumptions on the informativeness profile based weights $\omega_{x_{0},j}$. Simulations and real-life data study show competitive IMSE vs. full-data KRR and improved test MSE versus Nyström method and FALKON at comparable budgets.

**Strengths:**

The consitional MSE‑minimizing weights $w_{x,i,C}(x)\propto |K_{xi}|$ derived from the cluster‑center surrogate model are simple, somewhat interpretable, and targeted at the desired loss (pointwise prediction at a given $x$). Theorem 1 recovers the optimal full‑data KRR rate of convergenceunder different choices of target RKHS function class and eigendecay rate, while Theorem 2 formalizes how an informative sampling profile (captured by $\omega_{x_{0},j}$) yields an IMSE rate faster than SRS for fixed $n$. The paper shows the value of using information in the full input data to determine sub‑data selection.

**Weaknesses:**

Theorem 2 makes the assumption : $\omega_{x_0,j}\asymp j^{2\beta}$ with $0\le 2\beta<k$ (and $2\beta\le k\le 4\beta$). However, it is not shown that the proposed $k$‑means + $|K_x|$ weighting mechanism induces such a condition, for any $\beta>0$, even under standard kernels and benign marginal input disribution $\pi(x)$. As written, the theorem establishes rates for a class of informative samplers rather than for the concrete algorithm. A lemma connecting cluster geometry and $K_x$ to eigenfunction mass can strengthen the result. Also, using Euclidean $k$‑means to define centers may not align with the RKHS geometry for non‑RBF kernels. A short discussion (or a kernel‑$k$‑means variant such as $k$‑medians) would clarify when the centers faithfully represent $K(\cdot,\cdot)$. In the definition of Relative efficiency (RE) on Lines 316-318, the choice of the exponents of IMSE and Time is not well-motivated.

**Questions:**

Pleae see Weaknesses section.

---

> ### Author Response · Authors · 2025-11-20
>
> We greatly appreciate the valuable suggestions and constructive comments from the reviewers.
> We have devoted significant time and effort to carefully address each of them and have made
> revisions based on your comments and questions. For clarity, all major changes in the revised manuscript are highlighted in blue.
>
> 1. Clarification of Assumptions in Theorem 2
>
> Indeed, the proposed results are applicable to general sampling weights.
> Following your suggestions, we have included a lemma to understand the order of the weight
> functions $\omega_{x_0,j}$. Please see Lemma 1 in Section 3.2 and its proof in the Appendix.
> Roughly speaking, for most clustering methods such as K-means with reasonable cluster sizes
> and centers close to cluster means, the assumption $\omega_{x_0,j}\asymp j^\beta$ is justified
> if $\lambda_j\asymp j^{-k}$. Moreover, we have updated Theorem 2 for a more tight bound.
> See the updated results and proofs.
>
> 2. Alignment Between Clustering Centers and RKHS Geometry
>
> We thank the reviewer for raising this important point. When the underlying function $f_0$
> lies in an RKHS generated by a kernel $K$, a clustering algorithm using the distance function
> based on $K$ that respects the RKHS geometry, rather than relying on Euclidean distances,
> would be more appropriate than the standard Euclidean $k$-means. In the revised manuscript,
> we have added a brief discussion to clarify this point (see the second paragraph of
> Section 2.1). To further address your comments, we also included a numerical study in
> Section D of the Appendix to investigate the robustness of our method to clustering methods.
>
> 3. Justification of the Relative Efficiency(RE) Definition
>
> We sincerely appreciate the reviewer’s constructive suggestion. We chose
> $\alpha = 5/4$ and $\beta = 1/3$ for the definition of Efficiency so that the efficiency of
> the full-data method is (asymptotically) a constant (not varying) with respect to $N$,
> because Theorem 1 suggests that the full-data approach has $\mathrm{IMSE}\asymp N^{-4/5}$
> when $m=2$ (for functions in a Sobolev space of order $m$), and computation time
> $\asymp N^{3}$ (using direct computation). This choice also ensures that the RE of the
> method based on Simple Random Sampling is approximately 1. In the revised version, we added
> a short explanation in the second paragraph of Section 4.1.

---

### Meta-Review · Area_Chair_eehS · 2026-01-08

**Summary:**

This paper addresses the computational challenges of large-scale kernel learning by proposing an optimal sub-data selection method for pointwise nonparametric function estimation. The reviewers generally acknowledge the importance of the problem (scalability in kernel methods) and the theoretical efforts to establish convergence rates (Theorems 1 and 2). However, significant concerns were raised regarding the conceptual novelty of combining established techniques like k-means and variance minimization. More critically, the initial submission suffered from outdated baselines (e.g., FALKON 2017) and substantial presentation issues, including inconsistent notation and lack of foundational references.

**Reviewer Concerns:**

While the authors made extensive efforts during the rebuttal—including updating the experiments with more recent baselines (FALKON 2020), adding Lemma 1 to justify theoretical assumptions, and standardizing notations—several key concerns remain outstanding:

(1) Reviewers noted that the core components (k-means, variance minimization) are classical, and the proposed integration is seen as a recombination of existing techniques rather than a fundamental breakthrough.

(2) Despite the "optimal selection," the proposed method remains significantly slower (often by an order of magnitude or more) than state-of-the-art sketching methods like FALKON for bulk prediction. The authors' defense that it is optimized for "pointwise" prediction does not fully mitigate the practical scalability concerns for large datasets.

(3) Even after multiple revisions, some reviewers remained unsatisfied with the lingering notation errors and the overall clarity of the theoretical exposition.

**Reviewer Scores:**

Reviewer bZ3f: (Score: 6) Marginally above threshold. Likely would have maintained or slightly lowered the score due to the remaining concerns on the practical alignment of clustering with RKHS geometry.

Reviewer Xqhh: (Score: 2) Reject. Despite author updates, the fundamental criticism regarding a lack of conceptual innovation and poor initial quality likely persists

Reviewer dGCQ: (Score: 2) Reject. The concerns regarding "trivial findings" and the significant speed gap compared to modern baselines were only partially addressed by adding a "hybrid" method (Prop+FALKON)

Reviewer c2tu: (Score: 4) Marginally below threshold. Maintained the score even after rebuttal due to persistent issues in the meticulousness of the revision and notation.

---

### Decision · Program_Chairs · 2026-01-26

Reject